

**COMPUTO**

**ISSN 2824-7795**

# Draw Me a Simulator

## Using neural networks to build more realistic simulation schemes for causal analysis

Sandrine Boulet[1]    HeKA, UMRS 1346, F-75015 Paris, France, Inserm, Université Paris Cité, Inria

Antoine Chambaz ⓘ    MAP5, F-75006 Paris, France, Université Paris Cité, CNRS

Date published: 2025-03-21    Last modified: 2025-03-21

**Abstract**

This study investigates the use of Variational Auto-Encoders to build a simulator that approximates the law of genuine observations. Using both simulated and real data in scenarios involving counterfactuality, we discuss the general task of evaluating a simulator's quality, with a focus on comparisons of statistical properties and predictive performance. While the simulator built from simulated data shows minor discrepancies, the results with real data reveal more substantial challenges. Beyond the technical analysis, we reflect on the broader implications of simulator design, and consider its role in modeling reality.

*Keywords:* simulations, variational auto-encoders, counterfactuals

# Contents

---

[1]Corresponding author: sandrine.boulet@inria.fr

Asher Rubin walks out of the starosta's home and heads toward the market square. With
evening, the sky has cleared, and now a million stars are shining, but their light is cold
and brings down a frost upon the earth, upon Rohatyn. The first of this autumn. Rubin
pulls his black wool coat tighter around him; tall and thin, he looks like a vertical line.
(Tokarczuk 2021, I(3))

# 1 Introduction

## 1.1 Fiction as the original simulation

One of humanity's oldest creative endeavors, fiction represents an early form of simulation. It extends
the imaginative play where children create scenarios, roles, or worlds that are not constrained by
the rules of reality, that is "childhood pretence" (Carruthers 2002) or "the make-believe games" of
children (Walton 1993). Through stories, myths, and imagined worlds, humans construct alternative
realities to explore ideas, express emotions, and reflect on their existence. By presenting hypothetical
scenarios and posing "what if things had been different" questions (Pearl and Mackenzie 2018, 34),
fiction empowers individuals to explore alternative histories, draw insights from the experiences of
others, and engage with possibilities that extend beyond the confines of the physical world. At its core,
fiction abstracts and reconstructs elements of reality. An author selectively includes, exaggerates, or
omits aspects of the real world, creating models that serve their artistic or thematic intentions. From
Homer's *Odyssey* (Homère 2000) to speculative tales like Mary Shelley's *Frankenstein* (Shelley 1818),

fiction mirrors the complexities of human life, enabling readers to engaged with an imagined reality that resonates with their own.

The relationship between fiction and reality has long been a subject of debate. Plato, in his critique of art, viewed fiction as a mere imitation of the physical world, itself a flawed reflection of the ideal "Forms". By this reasoning, fiction is a "simulation of a simulation", twice removed from truth (Platon 2002, Livre X). Aristotle, by contrast, argued that fiction, through "mimesis", the imitation of action and life, can illuminate universal truths (Aristote 2006, Chapitres 1 à 5). By abstracting from the particular, fiction allows exploration of broader patterns and principles.

Following Aristotle's perspective, this tradition of creating and interacting with imagined realities provides a natural foundation for distinguishing scientific theories from scientific models (Barberousse and Ludwig 2000) and understanding modern simulations. While they stem from the same drive to represent and explore, scientific theories, scientific models and modern simulations introduce a higher degree of mathematical rigor. Nevertheless, fiction remains their conceptual ancestor, reminding us that the human impulse to model and engage with alternate realities is as old as storytelling itself.

## 1.2 From modern simulations to computer simulations

The concept of modern simulations predates the modern era. Early instances include mechanical devices like the Antikythera, a sophisticated analog computer from the 2nd century BCE designed to simulate celestial movements (and the MacGuffin chased by Indiana Jones in the 2024 installment of the franchise, Solly 2023). The emergence of mathematical models in the works of Galileo and Newton introduced a new form of simulation, where equations were used to predict physical phenomena with increasing precision. By the 18th century, probabilistic experiments like Buffon's Needle, designed to approximate the number $\pi$ (Aigner and Ziegler 2018, sec. 24), demonstrated the power of simulating complex systems. However, the advent of computer simulations, as we understand them today, began during World War II with the work of J. von Neumann and S. Ulam (Metropolis and Ulam 1949).

While studying neutron behavior, they faced a challenge that was too complex for theoretical analysis and too hazardous, time-consuming, and costly to investigate experimentally. Fundamental properties (e.g., possible events and their probabilities) and basic quantities (e.g., the average distance a neutron would travel before colliding with an atomic nucleus, the likelihood of absorption or reflection, and energy loss after collisions) were known, but predicting the outcomes of entire event sequences was infeasible. To address this challenge, they devised a method of generating random sequences step by step using a computer, naming it "Monte Carlo" after the casino, a suggestion by N. Metropolis. Statistical analysis of the data produced by repeatedly applying this method provided sufficiently accurate solutions to better understand nuclear chain reactions, a crucial aspect of designing atomic bombs and later nuclear reactors. This breakthrough marked the birth of modern computer simulations.

Today, computer simulations, henceforth referred to simply as *simulations*, play a fundamental role in applied mathematics. Generally, conducting a simulation involves running a computer program (a "simulator") designed to represent a "system of interest" at a problem-dependent level of abstraction (that is, with a specific degree of complexity) and collecting the numerical output for analysis.

Examples of systems of interest are virtually limitless and highly diverse. They can represent a real-world process in a holistic fashion, such as the regular functioning of a person's heart at rest, or the medical trajectories of a cohort of patients undergoing chemotherapy. Alternatively, in a more focused fashion, they can consist of a hybrid pipeline that combines an upstream real-world process with downstream data processing of intermediary outputs, such as the estimation of peripheral oxygen saturation in a healthy patient using a pulse oximeter. Regardless of the context, determining the appropriate levels of abstraction and realism is always a significant challenge.

Here, we focus on simulations used to evaluate the performance of statistical procedures through simulation studies, as discussed by Morris, White, and Crowther (2019) in their excellent tutorial on the design and conduct of such studies. The interested reader will find in their work a carefully curated list of books on simulation methods in general and articles emphasizing rigor in specific aspects of simulation studies. Specifically, we consider scenarios where a statistician, interested in a real-world process, has developed an algorithm tailored to learning a particular feature of that process from collected data and seeks to assess the algorithm's performance through simulations.

Once the simulator is devised, the following process is repeated multiple times. In each iteration, typically independently from previous iterations: first, the simulator generates a synthetic data set of size $n$; second, the algorithm is run on the generated data; third, the algorithm's output is collected for further analysis. After completing these iterations, the next step is to compare the outcome from one run to the algorithm's target. This is made possible by the design of the simulator. Finally, the overall performance of the algorithm is assessed by comparing all the results collectively to the algorithm's target. Depending on the task, this evaluation can involve assessing the algorithm's ability to well estimate its target, the validity of the confidence regions it constructs for its target, the algorithm's ability to detect whether its target lies within a specified null domain (using an alternative domain as a reference), and more. This list is far from exhaustive. The entire process can be repeated multiple times, for example, to assess how the algorithm's performance depends on $n$.

However, in order to carry out these steps, the statistician must first devise a simulator. This simulator should ideally generate synthetic data that resemble the real-world data in a meaningful way, a goal that is often difficult to achieve. So, how can one design a realistic simulator, and what does "realistic simulator" even mean in this context? These are the central questions we explore in this work.

## 1.3 A probabilistic stance

We adopt a probabilistic framework to model the data collected by observing a real-world process. Specifically, the data are represented as a random variable $O^n$ ($O$ as in *observations*) drawn from a probability law $P^n$ ($P$ as in *probability*). The law $P^n$ is assumed to belong to a statistical model $\mathcal{M}^n$ ($\mathcal{M}$ as in *model*), which is the set of all probability laws on the space $\mathcal{O}^n$ where $O^n$ takes its values. This model incorporates constraints that reflect known properties of the real-world process and, where necessary, minimal assumptions about it.

The superscript $n$ indicates an amount of information. For example, in the context of this study, $n$ typically represents the number of elementary observations drawn independently from a law $P$ on $\mathcal{O}$ and gathered in $O^n$. In this case, $\mathcal{O}^n$ corresponds to the Cartesian product $\mathcal{O} \times \cdots \times \mathcal{O}$ (repeated $n$ times) and $P^n$ to the product law $P^{\otimes n}$, with $O^n$ decomposing as $(O_1, \ldots, O_n)$

The feature of interest is an element of a space $\mathcal{F}$ (e.g., a subset of the real line, or a set of functions). It is modeled as the value $\Psi(P^n)$ of a functional $\Psi : \mathcal{M}^n \to \mathcal{F}$ evaluated at $P^n$. The algorithm developed to estimate this feature is modeled as a functional $\mathcal{A} : \mathcal{O}^n \to \mathcal{F}$. Training the algorithm involves applying $\mathcal{A}$ to the observed data $O^n$, resulting in the estimator $\mathcal{A}(O^n)$ for the estimand $\Psi(P^n)$.

We emphasize that we address the questions closing Section 1.2 without focusing on the specific nature of the functional of interest $\Psi$: how can one design a realistic simulator, and what does "realistic simulator" even mean in this context?

## 1.4 Draw me a simulator

When constructing simulators, there is a spectrum of approaches, varying in complexity and flexibility. At one end of the spectrum, simulators are built upon relatively simple parametric models. While these models are sometimes more elaborate, they often rely on standard forms or recurring techniques,

which streamlines their implementation. This approach is further reinforced by the common practice of using models proposed by others. Doing so not only saves effort but also facilitates meaningful comparisons between studies, as the same modeling framework is shared.

Regardless of the model's simplicity, parametric simulators are inherently limited and unable to capture the complexity of real-world processes. The term "unnatural" aptly describes this shortcoming, as these models are simplifications that abstract away many intricacies of reality. Even with sophisticated parametrizations, it is fundamentally impossible for such simulators to convincingly replicate the multifaceted interactions and variability inherent in "nature". Thus, parametric simulators, by their very essence, cannot achieve realism.

At the other end of the spectrum, one can also adopt a nonparametric approach through bootstrapping, which involves resampling data directly from the observed dataset. This method bypasses the need to specify a parametric model and instead leverages the structure of the real data to generate simulated samples.

Bootstrapping usually refers to a self-starting process that is supposed to continue or grow without external input. The term is sometimes attributed to the story where Baron Münchausen pulls himself and his horse out of a swamp by his pigtail, not by his bootstraps (Raspe 1866, chap. 4). In France, "bootstrap" is sometimes translated as "à la Cyrano", in reference to the literary hero Cyrano de Bergerac, who imagined reaching the moon by standing on a metal plate and repeatedly using a magnet to propel himself (Rostand 2005, Act III, Scene 13).

When dealing with independent and identically distributed (i.i.d.) samples, bootstrapping generates data that closely resemble the observed data. However, the origin of the term "bootstrapping" suggests a measure of incompleteness hence dissatisfaction, which is fitting in the context of this article. Indeed, a bootstrapped simulator can be viewed as both transparent and opaque, depending on the perspective. Conditionally on the real data, the simulator's behavior is transparent, as understanding it reduces to understanding the sampling mechanism over the set of indices $\{1, \ldots, n\}$. Unconditionally, however, one is again confronted with the limitation of knowledge about $P^n$, beyond recognizing it as an element of $\mathcal{M}^n$.

```
   ------------------------------------
 / I am a simulator. Press ENTER to run \
 \ the synthetic experiment.            /
   ------------------------------------
    \
     \

        __
       UooU\.'@@@@@@`.
       \__/(@@@@@@@@@@)
           (@@@@@@@@)
            `YY~~~~YY'
             ||    ||
```

In *Le Petit Prince* (de Saint-Exupéry 1943), the Little Prince dismisses the pilot's simple drawings of a sheep as unsatisfactory. Instead, he prefers a drawing of a box, imagining the perfect sheep inside.

```
/***************************************************************
 * I am a simulator. Press ENTER to run the synthetic experiment. *
 ***************************************************************/
```

Similarly, in simulations, straightforward simulators often fail to capture the complexity we seek, while black-box simulators, though opaque, can sometimes offer greater efficiency. Unlike the Little

Prince, however, we are not content with the box alone – we want to look inside, to understand and refine the mechanisms driving our simulator.

## 1.5 Organization of the article

In this article, we explore an avenue to build more realistic simulators by using real data and neural networks, more specifically, Variational Auto-Encoders (VAEs). To illustrate our approach, we focus on a simple example rooted in causal analysis, as the causal framework presents particularly interesting challenges.

Section 2 outlines our objectives and introduces a running example that serves as a unifying thread throughout the study. Section 3 provides a concise overview of VAEs, including their formal definition and the key ideas behind their training. Section 4 offers an explanation of how VAEs are constructed, while Section 5 presents a comprehensive implementation tailored to the running example. Using this VAE, Section 6 describes the construction of a simulator designed to approximate the law of simulated data and discusses methods for evaluating the simulator's performance. Section 7 extends this approach to a real-world dataset. Finally, Section 8 concludes the article with a literature review, a discussion of the challenges encountered, the limitations of the proposed approach, and some closing reflections.

*Note that the online version of this article is preferable to the PDF version, as it allows readers to directly view the code.* Throughout the article, we use a mix of `Python` (Van Rossum and Drake 2009) and `R` (R Core Team 2020) for implementation, leveraging commonly used libraries in both ecosystems.

## 2 Objective

Suppose that we have observed $O_1, \dots, O_n, O_{n+1}, \dots, O_{n+n'}$ drawn independently from $P$, with $P$ known to belong to a model $\mathscr{P}$ consisting of laws on $\mathcal{O}$. For brevity, we will use the notation $O^{1:n} = (O_1, \dots, O_n)$ and $O^{(n+1):(n+n')} = (O_{n+1}, \dots, O_{n+n'})$.

Suppose moreover that we are interested in a causal framework where each $O_i$ is viewed as a piece of a complete data $X_i \in \mathcal{X}$ drawn from a law $Q$ that lives in a model $\mathcal{Q}$, with $X_1, \dots, X_n, X_{n+1}, \dots, X_{n+n'}$ independent. The piece $O_i$ is expressed $\pi(X_i)$, with the function $\pi$ projecting a complete data $X \sim Q \in \mathcal{Q}$ onto a *coarser* real-world data $O = \pi(X) \sim P \in \mathscr{P}$.

Our objective is twofold. First, we aim to build a generator that approximates $P$, that is, an element of $\mathscr{P}$ from which it is possible to sample independent data that exhibit statistical properties similar to (or, colloquially, "behave like") $O_1, \dots, O_{n+n'}$. In other words, we require that the generator produces data whose joint law approximates the law of the observed data, ensuring that the generated samples reflect the same underlying structure and dependencies as the real-world observations. Second, we require the generator to correspond to the law of $\pi(X)$ with $X$ drawn from an element of $\mathcal{Q}$.

We use a running example throughout the document.

> **ℹ Running example.**
>
> For example, $\mathscr{P}$ can be the set of all laws on $\mathcal{O} := (\{0,1\}^2 \times \mathbb{R}^3) \times \{0,1\} \times \mathbb{R}$ such that
>
> $$O := (V, W, A, Y) \sim P \in \mathscr{P}$$
>
> satisfies
>
> $$c \leq P(A = 1|W, V), P(A = 0|W, V) \leq 1 - c$$
>
> $P$-almost surely for some $P$-specific constant $c \in ]0, 1/2]$, and $Y$ is $P$-integrable.
> Moreover, we view $O$ as $\pi(X)$ with
>
> $$X := (V, W, Y[0], Y[1], A) \in \mathscr{X} := (\{0,1\}^2 \times \mathbb{R}^3) \times \mathbb{R} \times \mathbb{R} \times \{0,1\},$$
> $$\pi : (v, w, y[0], y[1], a) \mapsto (v, w, a, ay[1] + (1 - a)y[0]),$$
>
> and $\mathcal{Q}$ defined as the set of all laws on $\mathscr{X}$ such that $X \sim Q \in \mathcal{Q}$ satisfies
>
> $$c' \leq Q(A = 1|W, V), Q(A = 0|W, V) \leq 1 - c'$$
>
> $Q$-almost surely for some $Q$-specific constant $c' \in ]0, 1/2]$, and $Y[0]$ and $Y[1]$ are $Q$-integrable.
> We consider $(V, W)$ as the context in which two possible actions $a = 0$ and $a = 1$ would yield
> the counterfactual rewards $Y[0]$ and $Y[1]$, respectively. One of these actions, $A \in \{0,1\}$, is
> factually carried out, resulting in the factual reward $Y = AY[1] + (1 - A)Y[0]$, that is, $Y[1]$ if
> $A = 1$ and $Y[0]$ otherwise. In the causal inference literature, this definition of $Y$ is referred
> to as the consistency assumption.

234

> **ℹ Running example in action.**
>
> The Python function `simulate` defined in the next chunk of code operationalizes drawing
> independent data from a law $P \in \mathcal{M}$.
>
> ```python
> import numpy as np
> import random
> from numpy import hstack, zeros, ones
>
> def simulate(n, dimV, dimW):
>   def expit(x):
>     return 1 / (1 + np.exp(-x))
>   p = np.hstack((1/3 * np.ones((n, 1)), 1/2 * np.ones((n, 1))))
>   V = np.random.binomial(n = 1, p = p)
>   W = np.random.normal(loc = 0, scale = 1, size = (n, dimW))
>   WV = np.hstack((W, V))
>   pAgivenWV = np.clip(expit(0.8 * WV[:, 0]), 1e-2, 1 - 1e-2)
>   A = np.random.binomial(n = 1, p = pAgivenWV)
>   meanYgivenAWV  = 0.5 * expit(-5 * A * (WV[:, 0] - 1)\
>                               + 3 * (1 - A) * (WV[:, 1] + 0.5))\
>                               + 0.5 * expit(WV[:, 2])
>   Y = np.random.normal(loc = meanYgivenAWV, scale = 1/25, size = n)
>   dataset = np.vstack((np.transpose(WV), A, Y))
>   dataset = np.transpose(dataset)
>   return dataset
> ```

235

Note that justifying the specific choices made while defining the function `simulate` is unnecessary. In the context of this study, we are free from the need for, or aspiration to, a realistic simulation scheme. Under the law $P$ that `simulate` samples from, $V$ and $W$ are independent; $V$ consists of two independent variables $V_1$ and $V_2$ that are drawn from the Bernoulli laws with parameters $\frac{1}{3}$ and $\frac{1}{2}$; $W$ is a standard Gaussian random variable. In addition, given $(W, V)$, $A$ is sampled from the Bernoulli law with parameter

$$\max\{0.01, \min[0.99, \text{expit}(0.8 \times W_1)]\}$$

and, given $(A, W, V)$, $Y$ is sampled from the Gaussian law with mean

$$\frac{1}{2}\text{expit}\left[-5A \times (W_1 - 1) + 3(1 - A) \times (\tfrac{1}{2} + W_2)\right] + \frac{1}{2}\text{expit}(W_3)$$

and (small) standard deviation $\frac{1}{25}$. As noted in the introduction, these choices rely on standard forms and recurring techniques.

> **ℹ Running example, cted.**
>
> For future use, we sample in the next chunk of code $n + n' = 10^4$ independent observations from $P$. Observations $O^{1:n}$ (gathered in `train`) will be used for training and observations $O^{(n+1):(n+n')}$ (gathered in `test`) will be used for testing.
>
> ```python
> import random
> random.seed(54321)
> dimV,  dimW = 2, 3
> n_train = int(5e3)
> train = simulate(n_train, dimV,  dimW)
> test = simulate(n_train, dimV,  dimW)
> print("The three first observations in 'train':\n",
>       "  V_1    V_2    W_1    W_2    W_3    A     Y\n",
>       np.around(train[:3, [3, 4, 0, 1, 2, 5, 6]], decimals = 3))
> The three first observations in 'train':
>    V_1    V_2    W_1    W_2    W_3    A     Y
>  [[ 1.    0.    0.568  1.982 -0.314  1.    0.694]
>  [ 0.    0.   -0.117  0.607  0.294  1.    0.82 ]
>  [ 1.    1.   -1.143 -0.71  -0.727  0.    0.279]]
> ## np.savetxt("data/train.csv", train, delimiter = ",")
> ## np.savetxt("data/test.csv", test, delimiter = ",")
> ```

## 3 VAE in a nutshell

### 3.1 Formal definition

In the context of this article, a Variational Auto-Encoder (VAE) (Kingma and Welling 2014), (Rezende, Mohamed, and Wierstra 2014) is an algorithm that, once trained, outputs a *generator*. The generator is the law of a random variable of the form

$$\text{Gen}_\theta(Z) \tag{1}$$

where

1. the source of randomness $Z$ in Equation 1 writes as

$$Z := (Z^{(0)}, \ldots, Z^{(d)}) \tag{2}$$

with $Z^{(0)}, \ldots, Z^{(d)}$ independently drawn from

- the uniform law on $\{1, \ldots, n\}$ for $Z^{(0)}$

- the standard normal distribution for $Z^{(1)}, \ldots, Z^{(d)}$;

2. the function $\mathrm{Gen}_\theta$ in Equation 1 is an element of a large collection, parametrized by the finite-dimensional set $\Theta$, of functions mapping $\mathbb{R}^{d+1}$ to $\mathcal{X}$.

Because $\mathrm{Gen}_\theta(Z)$ belongs to $\mathcal{X}$, we can evaluate $\pi \circ \mathrm{Gen}_\theta(Z)$, hence the generator can also be used to generate random variables in $\mathcal{O}$. Figure 1 illustrates the architecture of the VAE used in this study. It shows the key components of the model, including the encoder, the latent space, and the decoder, along with the flow of information between them.

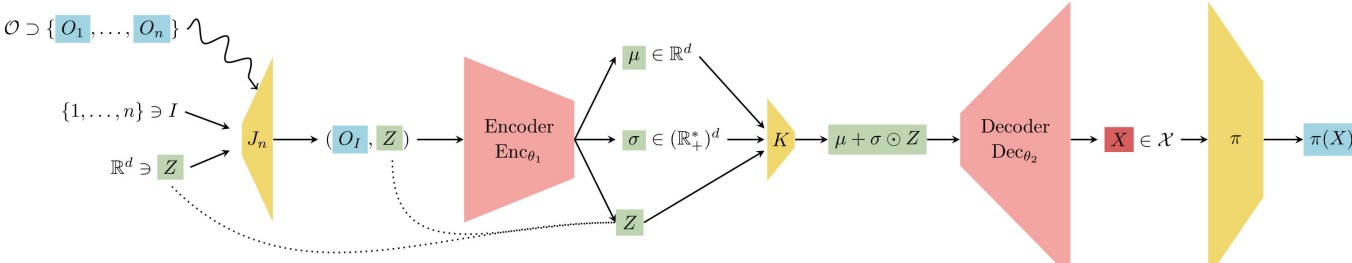

Figure 1: Architecture of the simulator. The figure depicts the flow of information through the encoder, latent space, and decoder components. It emphasizes how the input source of randomness $Z$ is transformed into a latent representation and then reconstructed as a complete data, $X = \mathrm{Gen}_\theta(Z)$, which can be mapped to a real-world data $O = \pi(X)$.

The word "auto-encoder" reflects the nature of the parametric form of each $\mathrm{Gen}_\theta$. We begin with a formal presentation in four steps, which is then followed by a discussion of what each step implements. Specifically, each $\mathrm{Gen}_\theta$ writes as a composition of four mappings $J_n$, $\mathrm{Enc}_{\theta_1}$, $K$ and $\mathrm{Dec}_{\theta_2}$ with $\theta := (\theta_1, \theta_2) \in \Theta_1 \times \Theta_2 = \Theta$:

$$\mathrm{Gen}_\theta = \mathrm{Dec}_{\theta_2} \circ K \circ \mathrm{Enc}_{\theta_1} \circ J_n.$$

Here,

1. $J_n : \{1, \ldots, n\} \times \mathbb{R}^d \to \mathcal{O} \times \mathbb{R}^d$ is such that

$$J_n(Z) = (O_i, (Z^{(1)}, \ldots, Z^{(d)}))$$

with $i = Z^{(0)}$;

2. $\mathrm{Enc}_{\theta_1} : \mathcal{O} \times \mathbb{R}^d \to \mathbb{R}^d \times (\mathbb{R}_+^*)^d \times \mathbb{R}^d$ is such that, if $\mathrm{Enc}_{\theta_1}(o, z) = (\mu, \sigma, z')$, then

- $z = z'$, and
- $\mathrm{Enc}_{\theta_1}(o, z'') = (\mu, \sigma, z'')$ for all $z'' \in \mathbb{R}^d$;

3. $K : \mathbb{R}^d \times (\mathbb{R}_+^*)^d \times \mathbb{R}^d \to \mathbb{R}^d$ is given by

$$K(\mu, \sigma, z) := \mu + \sigma \odot z,$$

where $\odot$ denotes the componentwise product;

4. $\mathrm{Dec}_{\theta_2}$ maps $\mathbb{R}^d$ to $\mathcal{X}$.

Conditionally on $O^{1:n}$ and $Z$, the computation of $\mathrm{Gen}_\theta(Z)$ is deterministic. The process unfolds in four steps:

1. **Sampling and transfer.** Compute $J_n(Z)$, which involves sampling one observation $O_i$ uniformly among all genuine observations and transfer $(Z^{(1)}, \dots, Z^{(d)})$ unchanged.

2. **Encoding step.** Compute $\mathrm{Enc}_{\theta_1} \circ J_n(Z)$, which encodes $O_i$ as a vector $\mu \in \mathbb{R}^d$ and a $d \times d$ covariance matrix $\mathrm{diag}(\sigma)^2$. This step does not modify $(Z^{(1)}, \dots, Z^{(d)})$, which is transferred unchanged.

3. **Gaussian sampling.** Compute $K \circ \mathrm{Enc}_{\theta_1} \circ J_n(Z)$ by evaluating $\mu + \sigma \odot (Z^{(1)}, \dots, Z^{(d)}) \in \mathbb{R}^d$. This amounts to sampling from the Gaussian law with mean $\mu$ and covariance matrix $\mathrm{diag}(\sigma)^2$.

4. **Decoding step.** Compute $\mathrm{Dec}_{\theta_2} \circ K \circ \mathrm{Enc}_{\theta_1} \circ J_n(Z)$, which maps the encoded version of $O_i$, that is, $\mu + \sigma \odot (Z^{(1)}, \dots, Z^{(d)})$, to an element of $\mathcal{X}$.

## 3.2 Formal training

Formally, training the VAE involves maximizing the likelihood of $O^{1:n}$ within a parametric model of laws by maximizing a lower bound of the likelihood. This process begins with the introduction of a working model of mixtures for $P$. The working model (undoubtedly flawed) postulates the existence of a latent random variable $U \in \mathbb{R}^d$ and a parametric model of *tractable* conditional densities

$$\{o \mapsto p_{\theta_2}(o|u) : u \in \mathbb{R}^d, \theta_2 \in \Theta_2\}$$

such that

- $U$ is drawn from the standard Gaussian law on $\mathbb{R}^d$;

- there exists $\theta_2 \in \Theta_2$ such that, given $U$, $O$ is drawn from $p_{\theta_2}(\cdot|U)$.

Here, tractable densities refer to those that can be easily worked with analytically, while in contrast, intractable densities are too complex to handle directly.

Therefore, the working model (undoubtedly flawed) postulates the existence of $\theta_2 \in \Theta_2$ such that $P$ admits the generally *intractable* density

$$o \mapsto \int p_{\theta_2}(o|u)\phi_d(u)du$$

where $\phi_d$ denotes the density of the standard Gaussian law on $\mathbb{R}^d$. As suggested by the use of the parameter $\theta_2$, the definition of the conditional densities $p_{\theta_2}(\cdot|u)$ ($u \in \mathbb{R}^d$) involves the decoder $\mathrm{Dec}_{\theta_2}$.

Since directly maximizing the likelihood of $O^{1:n}$ under the working model is infeasible, a *secondary* parametric model of *tractable* conditional densities is introduced:

$$\{u \mapsto g_{\theta_1}(u|O_i) : 1 \le i \le n, \theta_1 \in \Theta_1\}$$

to model the conditional laws of $U$ given $O_1$, given $O_2$, ..., given $O_n$. Here too, the use of the parameter $\theta_1$ indicates that the definition of the conditional densities $g_{\theta_1}(\cdot|O_i)$ ($1 \le i \le n$) involves the encoder $\mathrm{Enc}_{\theta_1}$.

Now, by Jensen's inequality, for any $1 \le i \le n$ and all $\theta = (\theta_1, \theta_2) \in \Theta$,

$$
\begin{aligned}
\log p_{\theta_2}(O_i) &= \log \int p_{\theta_2}(O_i|u) \frac{\phi_d(u)}{g_{\theta_1}(u|O_i)} g_{\theta_1}(u|O_i) du \\
&\ge \int \log \left( p_{\theta_2}(O_i|u) \frac{\phi_d(u)}{g_{\theta_1}(u|O_i)} \right) g_{\theta_1}(u|O_i) du \\
&= -\mathrm{KL}(g_{\theta_1}(\cdot|O_i); \phi_d) + E_{U \sim g_{\theta_1}(\cdot|O_i)}[\log p_{\theta_2}(O_i|U)] =: \underset{\theta}{\mathrm{LB}}(O_i),
\end{aligned}
\tag{3}
$$

where KL denotes the Kullback-Leibler divergence and $U$ in the expectation is drawn from the conditional law with density $g_{\theta_1}(\cdot|O_i)$. The notation LB is used to indicate that it represents a lower bound. Thus, the likelihood of $O^{1:n}$ under $\theta_2 \in \Theta$ is lower-bounded by

$$
\sum_{i=1}^{n} \underset{\theta}{\mathrm{LB}}(O_i)
$$

for all $\theta_1 \in \Theta_1$. As suggested earlier, training the VAE formally consists of solving

$$
\max_{\theta \in \Theta} \left\{ \sum_{i=1}^{n} \underset{\theta}{\mathrm{LB}}(O_i) \right\}
\tag{4}
$$

rather than solving

$$
\max_{\theta_2 \in \Theta_2} \left\{ \sum_{i=1}^{n} \log p_{\theta_2}(O_i) \right\}.
$$

# 4 How to build the VAE

## 4.1 A formal description

We implement the classes of encoders and decoders, that is $\{\mathrm{Enc}_{\theta_1} : \theta_1 \in \Theta_1\}$ and $\{\mathrm{Dec}_{\theta_2} : \theta_2 \in \Theta_2\}$, as neural network models. Each encoder $\mathrm{Enc}_{\theta_1}$ and decoder $\mathrm{Dec}_{\theta_2}$ consist of a stack of layers of two types: *densely-connected* and *activation* layers (linear, $x \mapsto x$; ReLU: $x \mapsto \max(0, x)$, softmax: $(x_1, x_2) \mapsto (e^{x_1}, e^{x_2})/(e^{x_1} + e^{x_2})$). The neural networks are rather simple in design, but nevertheless (moderately) high-dimensional and arguably over-parametrized, as discussed in Section 4.2.

The model $\{u \mapsto g_{\theta_1}(u|O_i) : 1 \le i \le n, \theta_1 \in \Theta_1\}$ is chosen such that $U$ drawn from $g_{\theta_1}(\cdot|O_i)$ is a Gaussian vector with mean $\mu_i$ and covariance matrix $\mathrm{diag}(\sigma_i)^2$ where $\mathrm{Enc}_{\theta_1}(O_i, \cdot) = (\mu_i, \sigma_i, \cdot)$, that is, when the $\theta_1$-specific encoding of $O_i$ equals $(\mu_i, \sigma_i)$. Remarkably, the left-hand side term in the definition of $\mathrm{LB}_{\theta}(O_i)$ (Equation 3) is then known in closed form:

$$
-\mathrm{KL}(g_{\theta_1}(\cdot|O_i); \phi_d) = \frac{1}{2} \sum_{j=1}^{d} \left( 1 + \log(\sigma_i^2)_j - (\sigma_i^2)_j - (\mu_i^2)_j \right),
\tag{5}
$$

where $(\mu_i^2)_j$ and $(\sigma_i^2)_j$ are the $j$-th components of $\mu_i \odot \mu_i$ and $\sigma_i \odot \sigma_i$, respectively. This is very convenient, because Equation 5 makes estimating the term $\text{KL}(g_{\theta_1}(\cdot|O_i); \phi_d)$ unnecessary, a task that would otherwise introduce more variability in the procedure.

As for the model $\{o \mapsto p_{\theta_2}(o|u) : u \in \mathbb{R}^d, \theta_2 \in \Theta_2\}$, the only requirement is that it must be chosen in such a way that $\log p_{\theta_2}(O_i|u)$ be computable for all $1 \le i \le n$, $\theta_2 \in \Theta_2$ and $u \in \mathbb{R}^d$. This is not a tall order as soon as $O$ can be decomposed as a sequence of (e.g., time-ordered) random variables that are vectors with categorical, or integer or real entries. Indeed, it then suffices (i) to decompose the likelihood accordingly under the form of a product of conditional likelihoods, and (ii) to choose a tractable parametric model for each factor in the decomposition. We illustrate the construction of $\{o \mapsto p_{\theta_2}(o|u) : u \in \mathbb{R}^d, \theta_2 \in \Theta_2\}$ in the context of our running example.

---

**ℹ Running example, cted.**

In the context of this example, $O = (V, W, A, Y)$ with $V \in \{0, 1\}^2$, $W \in \mathbb{R}^3$, $A \in \{0, 1\}$ and $Y \in \mathbb{R}$. Since the source of randomness $Z$ has dimension $(d + 1)$, $d$ must satisfy $d = d_1 + 3$ for some integer $d_1 \ge 1$.

Set $\theta = (\theta_1, \theta_2) \in \Theta$, $u \in \mathbb{R}^d$, and let $\pi \circ \text{Dec}_{\theta_2}(u) = (\tilde{v}, \tilde{w}, \tilde{a}, \tilde{y}) \in \mathcal{O}$. The conditional likelihood $p_{\theta_2}(O|u)$ (of $O$ given $U = u$) equals

$$p_{\theta_2}(V, W|u) \times p_{\theta_2}(A|W, V, u) \times p_{\theta_2}(Y|A, W, V, u)$$

so it suffices to define the conditional likelihoods $p_{\theta_2}(V, W|u)$ (of $(V, W)$ given $U = u$), $p_{\theta_2}(A|W, V, u)$ (of $A$ given $(W, V)$ and $U = u$) and $p_{\theta_2}(Y|A, W, V, u)$ (of $Y$ given $(A, W, V)$ and $U = u$).

- We decide that $V$ and $W$ are conditionally independent given $U$ under $p_{\theta_2}(\cdot|u)$. Therefore, it suffices to characterize the conditional likelihoods $p_{\theta_2}(V|u)$ (of $V$ given $U = u$) and $p_{\theta_2}(W|u)$ (of $W$ given $U = u$).
- We choose $w \mapsto p_{\theta_2}(w|u)$ to be the Gaussian density with mean $\tilde{w}$ and identity covariance matrix.

---

> **ℹ Running example, cted.**
>
> - The description of the conditional law of $V$ given $U = u$ under $p_{\theta_2}(\cdot|u)$ is slightly more involved. It requires that we give more details on the encoders and decoders.
>     - Like every encoder, $\mathrm{Enc}_{\theta_1}$ actually maps $\mathscr{O} \times \mathbb{R}^d$ to $[\mathbb{R}^{d_1} \times \{0\}^3] \times [(\mathbb{R}_+^*)^{d_1} \times \{1\}^3] \times \mathbb{R}^d$. In words, if $\mathrm{Enc}_{\theta_1}(o, \cdot) = (\mu, \sigma, \cdot)$, then it necessarily holds that the three last components of $\mu$ and $\sigma$ are 0 and 1, respectively. Therefore the three last components of the random vector $K \circ \mathrm{Enc}_{\theta_1} \circ J_n(Z)$ equal $Z^{(d-2)}, Z^{(d-1)}, Z^{(d)}$, three independent standard normal random variables.
>     - To compute $\mathrm{Dec}_{\theta_2}(u) = (\tilde{v}, \tilde{w}, \tilde{y}_0, \tilde{y}_1, \tilde{a}) \in \mathscr{X}$, we actually compute $\tilde{w}$ *then* $\tilde{v}$, then $(\tilde{y}_0, \tilde{y}_1, \tilde{a})$.
>         * The output $\tilde{w}$ is a $\theta_2$-specific deterministic function of the first $d_1$ components of $u$.
>         * The output $\tilde{v}$ is a $\theta_2$-specific deterministic function of the $(d_1 + 2)$ first components of $u$.
>           More specifically, two (latent) probabilities $\tilde{g}_1, \tilde{g}_2$ are first computed, as $\theta_2$-specific deterministic functions of the $d_1$ first components of $u$. Then $\tilde{v}_1$ and $\tilde{v}_2$ are set to $\mathbf{1}\{\Phi(u^{(d_1+1)}) \leq \tilde{g}_1\}$ and $\mathbf{1}\{\Phi(u^{(d_1+2)}) \leq \tilde{g}_2\}$, where $\Phi$ denotes the standard normal cumulative distribution function (c.d.f) and $u^{(d_1+1)}, u^{(d_1+2)}$ are the $(d_1 + 1)$-th and $(d_1 + 2)$-th components of $u$.
>           For instance, $\tilde{v}_1$ is given the value 1 if $\Phi(u^{(d_1+1)}) \leq \tilde{g}_1$ and 0 otherwise. Note that $\mathbf{1}\{\Phi(Z^{(d_1+1)}) \leq \tilde{g}_1\}$ follows the Bernoulli law with parameter $\tilde{g}_1$ because $Z^{(d_1+1)}$ is drawn from the standard normal law.
>         * The output $(\tilde{y}_0, \tilde{y}_1)$ is a $\theta_2$-specific deterministic function of $(\tilde{v}, \tilde{w})$ and the $d_1$ first components of $u$.
>         * The output $\tilde{a}$ is a $\theta_2$-specific deterministic function of $(\tilde{v}, \tilde{w})$ and the last component of $u$.
>           More specifically, a (latent) probability $\tilde{h}$ is first computed, as a $\theta_2$-specific deterministic function of $(\tilde{v}, \tilde{w})$. Then $\tilde{a}$ is set to $\mathbf{1}\{\Phi(u^{(d)}) \leq \tilde{h}\}$.
>           Note that $\mathbf{1}\{\Phi(Z^{(d)}) \leq \tilde{h}\}$ follows the Bernoulli law with parameter $\tilde{h}$ because $Z^{(d)}$ is drawn from the standard normal law.
>
>   We are now in a position to describe the conditional law of $V$ given $U = u$. We decide that, conditionally on $U = u$, under $p_{\theta_2}(\cdot|u)$, $V_1$ and $V_2$ are independently drawn from the Bernoulli laws with parameters $\tilde{g}_1$ and $\tilde{g}_2$. Thus, $p_{\theta_2}(\cdot|u)$ is such that $p_{\theta_2}(v|u) = [v_1\tilde{g}_1 + (1 - v_1)(1 - \tilde{g}_1)] \times [v_2\tilde{g}_2 + (1 - v_2)(1 - \tilde{g}_2)]$ for $v = (v_1, v_2) \in \{0, 1\}^2$.

> **ℹ Running example, cted.**
>
> - The description of the conditional law of $A$ given $(W, V)$ and $U = u$ under $p_{\theta_2}(\cdot|u)$ is similar to that of $V$ given $U$. We decide that, conditionally on $(W, V)$ and $U = u$, under $p_{\theta_2}(\cdot|W, V, u)$, $A$ follows the Bernoulli law with parameter $\underline{\tilde{h}}(V, W)$, where the probability $\underline{\tilde{h}}(v, w)$ lies between $\tilde{h}$ and $\bar{A}_n := \frac{1}{n}\sum_{i=1}^{n} A_i$ and is given, for any $(v, w) \in \{0, 1\}^2 \times \mathbb{R}^3$, by
>
> $$\underline{\tilde{h}}(v, w) := t(v, w)\tilde{h} + [1 - t(v, w)]\bar{A}_n \quad \text{with}$$
> $$-10\log t(v, w) = -\left[v_1 \log \tilde{g}_1 + (1 - v_1)\log(1 - \tilde{g}_1)\right]$$
> $$-\left[v_2 \log \tilde{g}_2 + (1 - v_2)\log(1 - \tilde{g}_2)\right]$$
> $$+ \|w - \tilde{w}\|_2^2.$$
>
> Thus, $p_{\theta_2}(\cdot|W, V, u)$ is such that $p_{\theta_2}(a|W, V, u) = a\underline{\tilde{h}}(V, W) + (1 - a)(1 - \underline{\tilde{h}}(V, W))$ for $a \in \{0, 1\}$.
> - Finally, we choose $y \mapsto p_{\theta_2}(y|A, W, V, u)$ to be the two-regime density given by
>
> $$p_{\theta_2}(y|A, W, V, u) = \frac{\mathbf{1}\{A = \tilde{a}\}}{\tilde{s}(W)}\phi_1\left(\frac{y - \tilde{y}}{\tilde{s}(W)}\right) + \mathbf{1}\{A \neq \tilde{a}\}C^{-1}$$
>
> where $\tilde{s}(w) := \frac{1}{\sqrt{5}}\|w - \tilde{w}\|_2$ for any $w \in \mathbb{R}^3$ and $C$ is the Lebesgue measure of the support of the marginal law of $Y$ under $P$ (it does not matter if $C$ is unknown). Thus, two cases arise:
>   - If $A = \tilde{a}$, then $Y$ is conditionally drawn under $p_{\theta_2}(\cdot|A, W, V, u)$ from the Gaussian law with mean $\tilde{y} = a\tilde{y}_1 + (1 - a)\tilde{y}_0$ and variance $\tilde{s}(W)^2$.
>   - Otherwise, $Y$ is conditionally drawn under $p_{\theta_2}(\cdot|A, W, V, u)$ from the uniform law on the support of the marginal law of $Y$ under $P$.
>
> Therefore, the conditional likelihood $p_{\theta_2}(Y|A, W, V, u)$ bears information only if $A = \tilde{a}$ (that is, if the actions $A$ and $\tilde{a}$ undertaken when generating $O = (V, W, A, Y)$ and computing $\mathrm{Dec}_{\theta_2}(u)$ coincide), which can be interpreted as a necessary condition to justify the comparison of the rewards $Y$ and $\tilde{y}$. Moreover, when $A = \tilde{a}$, the closer are the contexts $W$ and $\tilde{w}$, the more relevant is the comparison and the larger the magnitude of $p_{\theta_2}(Y|A, W, V, u)$ can be.

334

> **ℹ Running example, cted.**
>
> In summary, the right-hand side term in the definition of $\text{LB}_\theta(O_i)$ Equation 3 equals, up to a term that does not depend on $\theta$,
>
> $$\frac{1}{2}E_{U\sim g_{\theta_1}(\cdot|O_i)}\Bigg[ -2\left(V_{1,i}\log\tilde{G}_1 + (1 - V_{1,i})\log(1 - \tilde{G}_1)\right)$$
> $$-2\left(V_{2,i}\log\tilde{G}_2 + (1 - V_{2,i})\log(1 - \tilde{G}_2)\right)$$
> $$-\|W_i - \tilde{W}\|_2^2 \tag{6}$$
> $$-2\left(A_i\log\underline{\tilde{H}} + (1 - A_i)\log[1 - \underline{\tilde{H}}]\right)$$
> $$-\mathbf{1}\{A_i = \tilde{A}\} \times \left(\log\tilde{S}(W_i)^2 + \frac{(Y_i - \tilde{Y})^2}{2\tilde{S}(W_i)^2}\right)\Bigg],$$
>
> with the notational conventions $\pi \circ \text{Dec}_{\theta_2}(U) = (\tilde{V}, \tilde{W}, \tilde{A}, \tilde{Y})$, $V_i = (V_{i,1}, V_{i,2})$, and where $\tilde{G}_1$, $\tilde{G}_2, \underline{\tilde{H}}, \tilde{S}$ are defined like the above latent quantities $\tilde{g}_1, \tilde{g}_2, \underline{\tilde{h}}, \tilde{s}$ with $U$ substituted for $u$. The expression is easily interpreted: the opposite of Equation 6 is an average risk that measures - the likelihood of $V_{i,1}$ and $V_{i,2}$ from the points of view of the Bernoulli laws with parameters $\tilde{G}_1$ and $\tilde{G}_2$ (first and second terms), - the average proximity between $W_i$ and $\tilde{W}$ (third term), - the likelihood of $A_i$ from the point of view of the Bernoulli law with parameter $\underline{\tilde{H}}$ (fourth term), - the average proximity between $Y_i$ and $\tilde{Y}$ (fifth term) *only if* $A_i = \tilde{A}$ (otherwise, the comparison would be meaningless).
>
> In other terms, the opposite of Equation 6 can be interpreted as a measure of the average faithfulness of the reconstruction of $O_i$ under the form $\pi \circ \text{Dec}_{\theta_2}(U)$ with $U$ drawn from $g_{\theta_1}(\cdot|O_i)$. The larger is Equation 6, the better is the reconstruction of $O_i$ under the form $\pi \circ \text{Dec}_{\theta_2}(U)$ with $U$ drawn from $g_{\theta_1}(\cdot|O_i)$.
>
> To conclude, note that the conditional laws of $W$ and $Y$, both Gaussian, could easily be associated with diagonal covariance matrices different from the identity matrix. This adjustment would be particularly relevant in situations where $\|W\|_2$ and $|Y|$ are typically not of the same magnitude, with $O = (V, W, A, Y)$ drawn from the law $P$ of the experiment of interest. Alternatively, the genuine observations could be pre-processed to ensure that $\|W\|_2$ and $|Y|$ are brought to comparable magnitudes.

The hope is that, once the VAE is trained, yielding a parameter $\hat{\theta}_n = ((\hat{\theta}_n)_1, (\hat{\theta}_n)_2)$, the corresponding generator $\text{Gen}_{\hat{\theta}_n}$ produces a synthetic complete data $X \in \mathcal{X}$ such that the law of $\pi(X) \in \mathcal{O}$ closely approximates $P$. Naturally, this approximation is closely related to the conditional densities $g_{(\hat{\theta}_n)_1}(\cdot|O_i)$ and $p_{(\hat{\theta}_n)_2}(\cdot|u)$ ($1 \le i \le n, u \in \mathbb{R}$).

For instance, in the context of the running example, if $O = (V, W, A, Y) = \pi \circ \text{Gen}_{\hat{\theta}_n}(Z)$ and if $\xi, \zeta$ are independently drawn from the centered Gaussian laws with an identity covariance matrix on $\mathbb{R}^3$ and variance 1 on $\mathbb{R}$, respectively, then $(W + \xi, A, Y + \zeta)$ follows a law that admits the density

$$(v, w, a, y) \mapsto \int p_{(\hat{\theta}_n)_2}(y|a, w, v, u) \times \mathbf{1}\{a = \tilde{a}_{(\hat{\theta}_n)_2}(u)\}$$
$$\times\, p_{(\hat{\theta}_n)_2}(w, v|u)\left(\frac{1}{n}\sum_{i=1}^{n} g_{(\hat{\theta}_n)_1}(u|O_i)\right)du,$$

where $\tilde{a}_{(\hat{\theta}_n)_2}(u)$ is defined as the $A$-coefficient of $\pi \circ \text{Dec}_{(\hat{\theta}_n)_2}(u)$.

## 4.2 About the over-parametrization

In Section 4.1 we acknowledged that the models $\{\text{Enc}_{\theta_1} : \theta_1 \in \Theta_1\}$ and $\{\text{Dec}_{\theta_2} : \theta_2 \in \Theta_2\}$ are over-parametrized in the sense that the dimensions of the parameter set $\Theta_1 \times \Theta_2$ is potentially large. For instance, the dimension of the model that we build in the next section is 1157. This is a common feature of neural networks.

Our models are also over-parametrized in the sense that they are not identifiable. This is obviously the case because of the loss of information that governs the derivation of an observation $O$ as a piece $\pi(X)$ of a complete data $X$ that we are not given to observe in its entirety.

> **i    Running example, cted.**
>
> In particular, in the context of this example, it is well know that we cannot learn from $O_1, \dots, O_n$ any feature of the joint law of the counterfactual random variables $(Y[0], Y[1])$ that does not reduce to a feature of the marginal laws of $Y[0]$ or $Y[1]$, unless we make very strong assumptions on this joint law (e.g., that $Y[0]$ and $Y[1]$ are independent).

This is not a source of concern. First, it is generally recognized that the fitting of neural networks often benefits from the high dimensionality of the optimization space and the presence of numerous equivalently good local optima, resulting in a redundant optimization landscape (Choromanska et al. 2015), (Arora, Cohen, and Hazan 2018). Second, our objective is to construct a generator that approximates the law $P$ of $O_1, \dots, O_n$, generating $O \in \mathcal{O}$ by first producing $X \in \mathcal{X}$ (via $\text{Gen}_\theta(Z)$) and then providing $\pi(X)$. The fact that two different generators $\text{Gen}_\theta$ and $\text{Gen}_{\theta'}$ can perform equally well is not problematic. Identifying *one* generator $\text{Gen}_\theta$ that performs well is sufficient.

It is possible to search for generators that satisfy user-supplied constraints, provided these can be expressed as a real-valued criterion $F(E[\mathscr{C}(\text{Gen}_\theta(Z))])$. For example, one may wish to construct a generator $\text{Gen}_\theta$ such that the components of $X$ under $\text{law}(\text{Gen}_\theta)$ exhibit a pre-specified correlation pattern (as demonstrated in the simple example below).

To focus the optimization procedure on generators that approximately meet these constraints, one can modify the original criterion Equation 4 by adding a penalty term. Specifically, given a user-supplied hyper-parameter $\lambda > 0$, we can substitute

$$\max_{\theta \in \Theta} \left\{ \sum_{i=1}^{n} \underset{\theta}{\text{LB}}(O_i) + \lambda F(E_{Z \sim \text{Unif}\{1,\dots,n\} \otimes N(0,1)^{\otimes d}}[\mathscr{C}(\underset{\theta}{\text{Gen}}(Z))]) \right\} \tag{7}$$

for Equation 4. From a computational perspective, this adjustment simply involves adding the term

$$\lambda F\left( \frac{1}{m} \sum_{i=1}^{m} \mathscr{C}(\underset{\theta}{\text{Gen}}(Z_{m+i})) \right) \tag{8}$$

to the expressions within the curly brackets in the definition of $g$ in the algorithm described in Section 5.5.

> **ℹ Running example, cted.**
>
> In particular, in the context of this example, we could look for generators $\text{Gen}_\theta$ such that the correlation of $Y[0]$ and $Y[1]$ under $\text{law}(\text{Gen}_\theta(Z))$ be close to a target correlation $r \in\ ]-1, 1[$. In that case, we could choose $\mathscr{C}(\text{Gen}_\theta(Z)) := (Y[0]Y[1], Y[0]^2, Y[1]^2, Y[0], Y[1])$ and $F : (a, b, c, d, e) \mapsto |(a - de)/\sqrt{(b - d^2)(c - e^2)} - r|$.

# 5 Implementation of the VAE in the context of the running example

We now show how to implement the classes of encoders and decoders, hence of generators, in the context of our running example. We will also define other loss functions that are needed to train the model.

We implemented our approach using the TensorFlow package, but also experimented with PyTorch. Both frameworks yielded similar results, with no noticeable differences in performance. However, we found TensorFlow to be slightly more beginner-friendly, which might make it easier for readers new to neural network frameworks to follow our implementation.

## 5.1 Implementing the encoder

The first chunk of code defines a function, namely `build_encoder`, to build $\text{Enc}_{\theta_1}$. The parameter `latent_dim` is the Python counterpart of $d_1$. The parameters `nlayers_encoder` and `nneurons_encoder` are the numbers of layers and of neurons in each layer, respectively. The parameter `L` will be discussed later.

```
Traceback (most recent call last):
  File "/home/runner/micromamba/envs/micromamba/lib/python3.12/site-packages/tensorflow/python/pyw
    import ssl
  File "/home/runner/work/draw_me_a_simulator/draw_me_a_simulator/renv/cache/v5/linux-
ubuntu-noble/R-4.4/x86_64-pc-linux-gnu/reticulate/1.41.0.1/43239d1c5749802890d904295dbdc4a8/reticu
    return _run_hook(name, _hook)
           ^^^^^^^^^^^^^^^^^^^^^^^
  File "/home/runner/work/draw_me_a_simulator/draw_me_a_simulator/renv/cache/v5/linux-
ubuntu-noble/R-4.4/x86_64-pc-linux-gnu/reticulate/1.41.0.1/43239d1c5749802890d904295dbdc4a8/reticu
    module = hook()
             ^^^^^^
  File "/home/runner/work/draw_me_a_simulator/draw_me_a_simulator/renv/cache/v5/linux-
ubuntu-noble/R-4.4/x86_64-pc-linux-gnu/reticulate/1.41.0.1/43239d1c5749802890d904295dbdc4a8/reticu
    return _find_and_load(name, import_)
           ^^^^^^^^^^^^^^^^^^^^^^^^^^^^^^^
  File "/home/runner/micromamba/envs/micromamba/lib/python3.12/ssl.py", line 100, in <module>
    import _ssl             # if we can't import it, let the error propagate
    ^^^^^^^^^^^^
  File "/home/runner/work/draw_me_a_simulator/draw_me_a_simulator/renv/cache/v5/linux-
ubuntu-noble/R-4.4/x86_64-pc-linux-gnu/reticulate/1.41.0.1/43239d1c5749802890d904295dbdc4a8/reticu
    return _run_hook(name, _hook)
           ^^^^^^^^^^^^^^^^^^^^^^^
  File "/home/runner/work/draw_me_a_simulator/draw_me_a_simulator/renv/cache/v5/linux-
ubuntu-noble/R-4.4/x86_64-pc-linux-gnu/reticulate/1.41.0.1/43239d1c5749802890d904295dbdc4a8/reticu
    module = hook()
```

409            ^^^^^^
410    File "/home/runner/work/draw_me_a_simulator/draw_me_a_simulator/renv/cache/v5/linux-
411 ubuntu-noble/R-4.4/x86_64-pc-linux-gnu/reticulate/1.41.0.1/43239d1c5749802890d904295dbdc4a8/reticu
412      return _find_and_load(name, import_)
413             ^^^^^^^^^^^^^^^^^^^^^^^^^^^^^
414 ImportError: /usr/lib/x86_64-linux-gnu/libcrypto.so.3: version `OPENSSL_3.3.0' not found (required
415 dynload/_ssl.cpython-312-x86_64-linux-gnu.so)
416
417
418 Warning: Failed to load ssl module. Continuing without ssl support.

419 The code related to encoding is complete.

## 5.2 Implementing the decoder

421 The first chunk of code defines the component of $\mathrm{Dec}_{\theta_2}$, namely `build_WV_decoder`, that generates
422 $(V, W)$ based on $U$. It also defines a function, `as_sample`, that allows to *approximately* draw from
423 a discrete distribution. The parameters `nlayers_WV_decoder` and `nneurons_WV_decoder` are the
424 numbers of layers and of neurons in each layer, respectively. The parameter `L` will be discussed later.

425 We say that `as_sample` allows to sample *approximately* from a discrete distribution since we cannot
426 simply draw from it because of the need for this operation to be differentiable with respect to (w.r.t.)
427 the parameters of the neural network. Instead, we use the fact that, for $\beta > 0$ a large constant and $Z$
428 a standard normal random variable, the law of the random variable

$$\mathrm{expit}(-\beta(\Phi(Z) - p)) \tag{9}$$

429 (recall that $\Phi$ is the c.d.f. of the standard normal law) is concentrated around $\{0, 1\}$, a small neighbor-
430 hood of 1 having mass approximately $p$ and a small neighborhood of 0 having mass approximately
431 $(1 - p)$. For instance Figure 2 shows the empirical cumulative distribution function of 1000 indepen-
432 dent copies of the random variable defined in Equation 9 with $\beta = 30$ and $p = 1/3$:

433 The second chunk of code defines the component of $\mathrm{Dec}_{\theta_2}$, namely `build_Alaw_decoder`, that
434 generates a conditional law for $A$ given $(V, W)$. The parameters `nlayers_Alaw_decoder` and
435 `nneurons_Alaw_decoder` are the numbers of layers and of neurons in each layer, respectively. The
436 parameter `L` will be discussed later.

437 The third chunk of code defines the component of $\mathrm{Dec}_{\theta_2}$, namely `build_AYaY_decoder`, that generates
438 the counterfactual outcomes $Y[0]$ and $Y[1]$, the action carried out $A$ and the corresponding reward
439 $Y$. The parameters `nlayers_AYaY_decoder` and `nneurons_AYaY_decoder` are the numbers of layers
440 and of neurons in each layer, respectively. The parameter `L` will be discussed later.

441 Two comments are in order:

442 • Given the counterfactual rewards $Y[0]$ and $Y[1]$ (outputs of the layer `'Ya'` in `build_AYaY_decoder`),
443   given the approximate action $A^\flat$ (output of the layer `'as_A'` in `build_AYaY_decoder`), the
444   actual reward $Y$ (output of the layer `'Y'` in `build_AYaY_decoder`) is defined as the weighted
445   mean

$$Y = A^\flat Y[1] + (1 - A^\flat)Y[0]$$

446   with $A^\flat$ close to 0 and 1 (see the above comment on `as_sample`).

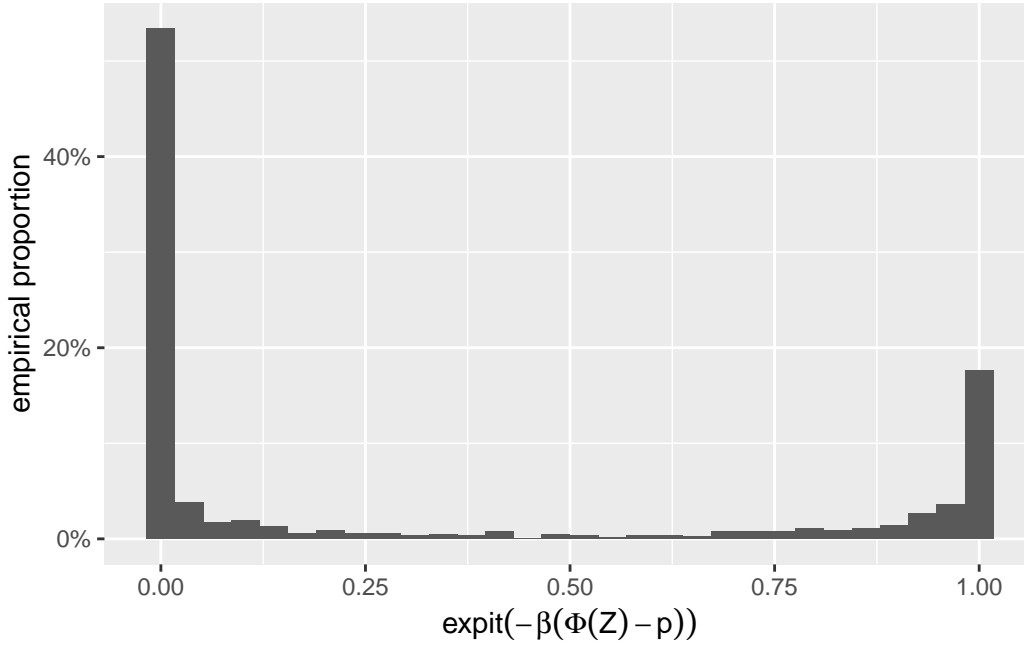

Figure 2: Empirical c.d.f. of 1000 independent copies of the random variable defined in Equation 9 with $\beta = 30$ and $p = 1/3$. The law is close the Bernoulli law with parameter $\frac{1}{3}$.

- The actual action $A$ (output of the layer `lambda_A` in `build_AYaY_decoder`) is derived from $A^\flat$ under the form

$$A = \frac{\text{ReLU}\left(A^\flat - \frac{1}{2}\right)}{A^\flat - \frac{1}{2}},$$

assuming that $A^\flat$ never takes on the value $\frac{1}{2}$. By doing so, $A$ is (almost everywhere) differentiable w.r.t. the parameters of the neural network.

The code related to decoding is complete.

## 5.3 Implementing the coarsening functions

The first chunk of code defines a function used to build the coarsening function $\pi$.

The next chunk of code defines a function used to extract the conditional probability that $A = 1$ given $W$ (denoted earlier as $\tilde{G}$).

## 5.4 Implementing the generator

At long last we are in a position to define a function, namely `build_generator`, whose purpose is to build the generator $\text{Gen}_\theta$. The chunk of code also defines the function `K` which is the Python counterpart of $K$ introduced in Section 3.1 and Section 4.1.

## 5.5 Implementing the loss functions and training algorithm

The last step of the encoding consists of defining the loss functions and optimization algorithm to drive the training of the VAE involved in Equation 6 by solving Equation 7. The definitions of the loss functions follow straightforwardly from the equations. As for the optimization algorithm, we rely on the minibatch stochastic ascent algorithm presented below:

**Algorithm 1** Minibatch stochastic gradient ascent training.

---

**Require:** number of epochs EPOCH, batch size $m$, number of repetitions $L$, learning rate $\alpha$, exponential decay rates $\beta_1$ and $\beta_2$ for the 1st and 2nd moments estimates, small constant $\epsilon$, initial parameter $\theta^{(0)} = (\theta_1^{(0)}, \theta_2^{(0)}) \in \Theta$

1: Initialize $D \leftarrow \{O_1, \dots, O_n\}$
2: Initialize $t \leftarrow 0$, $\text{first}^{(0)} \leftarrow 0_{\mathbb{R}^{\dim(\Theta)}}$, $\text{second}^{(0)} \leftarrow 0_{\mathbb{R}^{\dim(\Theta)}}$
3: **while** $t < \text{EPOCH}$ **do**
4:    Sample uniformly without replacement a minibatch of $m$ genuine observations $\tilde{O}_1, \dots, \tilde{O}_m$ from $D$
5:    Sample a minibatch of $m \times L$ independent sources of randomness $Z_{1,1}, \dots, Z_{1,L}, Z_{2,1}, \dots, Z_{2,L}, \dots, Z_{m,1}, \dots, Z_{m,L}$ from $(\mathcal{N}(0,1))^{\otimes d}$
6:    **for** $i = 1, \cdots, m$ **do**
7:        Compute $\text{Enc}_{\theta_1^{(t)}}(\tilde{O}_i, Z_{1,1}) = ((\mu_i)^{(t)}, (\sigma_i^2)^{(t)}, Z_{1,1})$
8:        **for** $\ell = 1, \cdots, L$ **do**
9:            $U_{i,\ell} \leftarrow (\mu_i)^{(t)} + \sqrt{(\sigma_i^2)^{(t)}} \odot (Z_{i,\ell}^{(1)}, \cdots, Z_{i,\ell}^{(d)})$
10:        **end for**
11:    **end for**
12:    Update the encoder and decoder by performing one step of stochastic gradient ascent:
13: $g \leftarrow \nabla_\theta \left\{ \frac{1}{m} \sum_{i=1}^m \left( -\text{KL}(g_{\theta_1}(\cdot|\tilde{O}_i); \phi_d) + \frac{1}{L} \sum_{\ell=1}^L \log p_{\theta_2}(\tilde{O}_i|U_{i,\ell}) \right) \right\} \Big|_{\theta=\theta^{(t)}}$
14: where, for each $1 \leq i \leq m$,
15: $-\text{KL}(g_{\theta_1^{(t)}}(\cdot|\tilde{O}_i); \phi_d) = \frac{1}{2} \sum_{j=1}^d \left( 1 + \log(\sigma_i^2)_j^{(t)} - (\sigma_i^2)_j^{(t)} - [(\mu_i)_j^{(t)}]^2 \right)$
16: $\text{first}^{(t+1)} \leftarrow \beta_1 \text{first}^{(t)} + (1-\beta_1)g$
17: $\text{second}^{(t+1)} \leftarrow \beta_2 \text{second}^{(t)} + (1-\beta_2)g \odot g$
18: $\widehat{\text{first}}^{(t+1)} \leftarrow \dfrac{\text{first}^{(t)}}{1-\beta_1^{t+1}}$
19: $\widehat{\text{second}}^{(t+1)} \leftarrow \dfrac{\text{second}^{(t)}}{1-\beta_2^{t+1}}$
20: $\theta^{(t+1)} \leftarrow \theta^{(t)} + \alpha \dfrac{\widehat{\text{first}}^{(t+1)}}{\sqrt{\widehat{\text{second}}^{(t+1)}} + \epsilon}$
21:
22:    Update $t \leftarrow t + 1$
23: **end while**

---

In our experiments, we set $\text{EPOCH} = 10$, $m = 10^3$, $L = 8$, $\alpha = 0.01$, $\beta_1 = 0.9$, $\beta_2 = 0.999$, $\epsilon = 10^{-7}$. The value of $L$ is chosen to be small for computational efficiency and to help the algorithm avoid getting stuck in local minima. The initial parameter $\theta^{(0)}$ is drawn randomly as follows: each component corresponding to a bias term in a densely-connected layer is set to 0; each component corresponding to a kernel coefficient is drawn independently of the others from the Glorot uniform initializer (Glorot and Bengio 2010) (that is, from the uniform law on $\sqrt{6/\ell} \times [-1, 1]$ where $\ell$ is the sum of the number of input units in the weight tensor and of the number of output units).

The next chunk of code defines the loss functions, optimization algorithm, and the `VAE` class which wraps up the implementation. The so-called `penalization_loss` is the counterpart of Equation 8.

## 6  Illustration on simulated data

In Section 2, in the context of the running example, we define a simulation law $P$ and simulated from $P$ a training data set train and a testing data set test using the function simulate. The two independent data sets consist of $n = 5000$ mutually independent realizations $O_i = (V_i, W_i, A_i, Y_i) \in \mathcal{O}$. We present here how to use train and the VAE coded in Section 5 to learn a function $\mathrm{Gen}_\theta$ so that, if $Z$ is sampled as in Equation 2, then $\mathrm{Gen}_\theta(Z)$ is a random element of $\mathcal{X}$ and $\pi \circ \mathrm{Gen}_\theta(Z)$ is a random element of $\mathcal{O}$ whose law approximates $P$.

### 6.1  Training the VAE

By running the next chunk of code, we set the VAE's configuration.

The next chunk of code repeatedly generates and initializes a VAE then trains it.

Because running the chunk is time-consuming, we stored one trained VAE that we considered good enough. We explain what we mean by good enough in the next section.

### 6.2  A formal view on how to evaluate the quality of the generator

Suppose that we have built a generator $\mathrm{Gen}_{\hat{\theta}_n}$ based on the genuine observations $O_1, ..., O_n$ drawn from $P$. How can we assess how well the generator approximates $P$? In other words, how can we assess how convincing are synthetic observations drawn from $\mathrm{law}(\mathrm{Gen}_{\hat{\theta}_n}(Z))$ in their attempt to look like observations drawn from $P$?

We propose three ways to address this question. Each of them uses the genuine observations $O_{n+1}, ..., O_{n+n'}$ that were not used to build $\mathrm{Gen}_{\hat{\theta}_n}$ and $N$ synthetic observations $O_1^\sharp, ..., O_N^\sharp$ drawn independently from $\mathrm{Gen}_{\hat{\theta}_n}$.

#### 6.2.1  Criterion 1

The overly faithful replication (a form of overfitting) by $\mathrm{Gen}_{\hat{\theta}_n}$ of $O_1, ..., O_n$, the genuine observations upon which its construction is based, is a pitfall that we aim to avoid. As a side note, the simplest generator that one can build from $O_1, ..., O_n$ is the empirical measure based on them, which corresponds to the bootstrap approach (see Section 1.4).

The first criterion we propose is inspired by a commonly used machine learning metric for comparing synthetic images generated by a neural network to the original training images. To assess the potential over-faithfulness of the replication process, we suggest comparing two empirical distributions:

- $\mu_{1:n}$, the empirical law of the distance to the nearest neighbor within $\{O_1^\sharp, ..., O_N^\sharp\}$ of each $O_i$ $(1 \le i \le n)$;
- $\mu_{(n+1):(n+n')}$, the empirical law of the distance to the nearest neighbor within $\{O_1^\sharp, ..., O_N^\sharp\}$ of each $O_{n+i}$ $(1 \le i \le n')$.

Ideally, $\mu_{1:n}$ and $\mu_{(n+1):(n+n')}$ should be similar, indicating that the training and testing performances align well. However, if $\mathrm{Gen}_{\hat{\theta}_n}$ replicates $O_1, ..., O_n$ too faithfully, then $\mu_{1:n}$ will become very concentrated around 0 while $\mu_{(n+1):(n+n')}$ will not exhibit the same behavior. Note that within a bootstrap approach, the generator that merely samples uniformly from $\{O_1, ..., O_n\}$ would result in a $\mu_{1:n}$ having all its mass at 0 if we let $N$ go to infinity, according to the law of large numbers.

### 6.2.2 Criterion 2

The second criterion involves comparing the marginal distributions of each real-valued component of $O$ under sampling from $P$ and from $\mathrm{law}(\mathrm{Gen}_{\hat{\theta}_n}(Z))$. This comparison can be conducted visually, by plotting the empirical distribution functions, or numerically, by computing $p$-values of hypotheses tests. Depending on the nature of the components of $O$, appropriate tests include the binomial, multinomial, $\chi^2$ or Kolmogorov-Smirnov tests.

### 6.2.3 Criterion 3

The third criterion aims to capture discrepancies between $P$ and $\mathrm{law}(\mathrm{Gen}_{\hat{\theta}_n}(Z))$ beyond marginal comparisons. To do so in general we propose, for a user-specified collection of prediction algorithms $\mathscr{A}_1, \dots, \mathscr{A}_K$, to compare their outputs when trained on $\{O_1, \dots, O_n\}$ versus $\{O_1^\sharp, \dots, O_n^\sharp\}$, using the predictions they make for each $O_{n+1}, \dots, O_{n+n'}$.

For instance, $\mathscr{A}_1$ could be an algorithm that learns to predict $A$ given $(V, W)$ based on the logistic regression model

$$\{(v, w) \mapsto m_\gamma(v, w) := \mathrm{expit}(\gamma^0 + \gamma^1(v, w)) : \gamma = (\gamma^0, \gamma^1) \in \mathbb{R} \times \mathbb{R}^5\}.$$

Training $\mathscr{A}_1$ on $\{O_1, \dots, O_n\}$ (respectively, $\{O_1^\sharp, \dots, O_n^\sharp\}$) yields $\gamma_n$ (respectively, $\gamma_n^\sharp$), hence the predictions $m_{\gamma_n}(V_{n+i}, W_{n+i})$ and $m_{\gamma_n^\sharp}(V_{n+i}, W_{n+i})$ $(1 \le i \le n)$. The closer $\mathrm{Gen}_{\hat{\theta}_n}$ approximates $P$, the nearer the points $(m_{\gamma_n}(V_{n+1}, W_{n+1}), m_{\gamma_n^\sharp}(V_{n+1}, W_{n+1})), \dots, (m_{\gamma_n}(V_{n+n'}, W_{n+n'}), m_{\gamma_n^\sharp}(V_{n+n'}, W_{n+n'}))$ are to the $y = x$ line in the $xy$-plane.

Importantly, the algorithms need not rely on parametric working models. For instance, $\mathscr{A}_2$ could learn to predict $A$ given $(V, W)$ using a nonparametric algorithm such as a random forest.

## 6.3 Implementing an evaluation of the quality of the generator

We now show how to implement the three criteria presented in Section 6.2. The next chunk of code loads the data into R: `train` and `test` are the R counterparts of the Python objects `train` and `test` (keeping only the first 1000 observations) and `synth` is the collection of 1000 synthetic observations drawn from the generator associated to the VAE that we stored in Section 6.1. For later use (while implementing Criterion 1) we add a dummy column named Z.

### 6.3.1 Criterion 1

The next chunk of code implements the first criterion.

The two empirical c.d.f. shown in Figure 3 are quite similar, suggesting that $\mu_{1:n}$ and $\mu_{(n+1):(n+n')}$ are close. To quantify this proximity, we rely on statistical tests.

The Directed Acyclic Graph (DAG) in Figure 4 represents the experiment of law $\Pi$ that consists successively of

- drawing $O_1, \dots, O_n, O_{n+1}, \dots, O_{n+n'}$ independently from $P$;
- learning $\hat{\theta}_n$;
- sampling $O_1^\sharp, \dots, O_N^\sharp$ independently from $\mathrm{law}(\mathrm{Gen}_{\hat{\theta}_n}(Z))$;
- determining, for each $1 \le i \le n + n'$, the nearest neighbor $f^\sharp(O_i)$ of $O_i$ among $O_1^\sharp, \dots, O_N^\sharp$.

The DAG is very useful to unravel how the random variables produced by $\Pi$ depend on each other. In particular, by $d$-separation (Lauritzen 1996), we learn from the DAG that the distances to the nearest

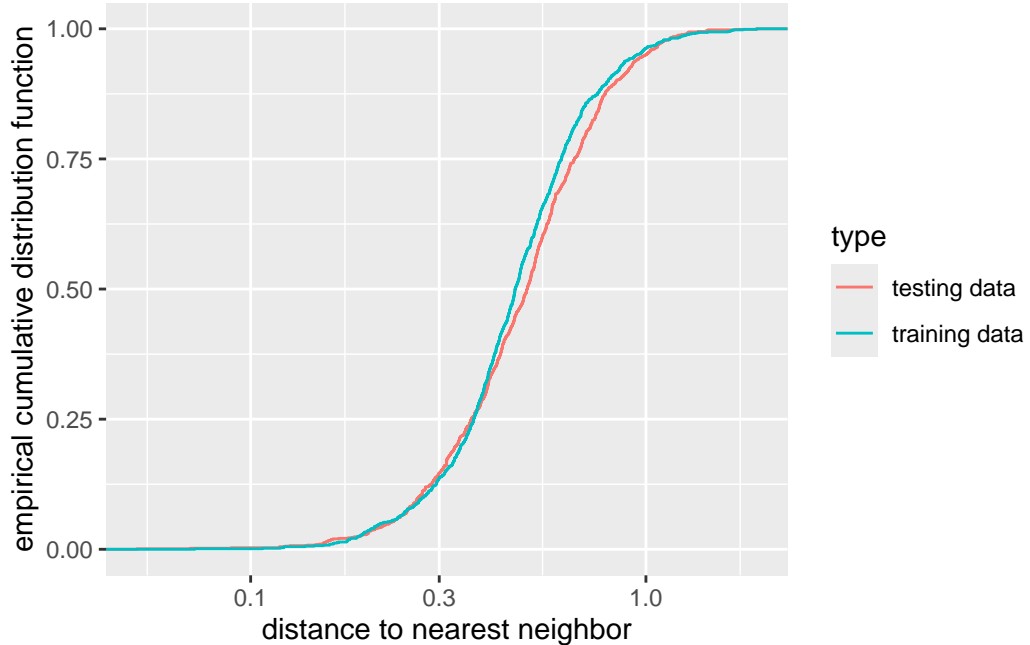

Figure 3: Empirical c.d.f. of the distance to the nearest neighbor within the synthetic observations of the training and of the testing data points (logarithmic scale). The two c.d.f. are quite close.

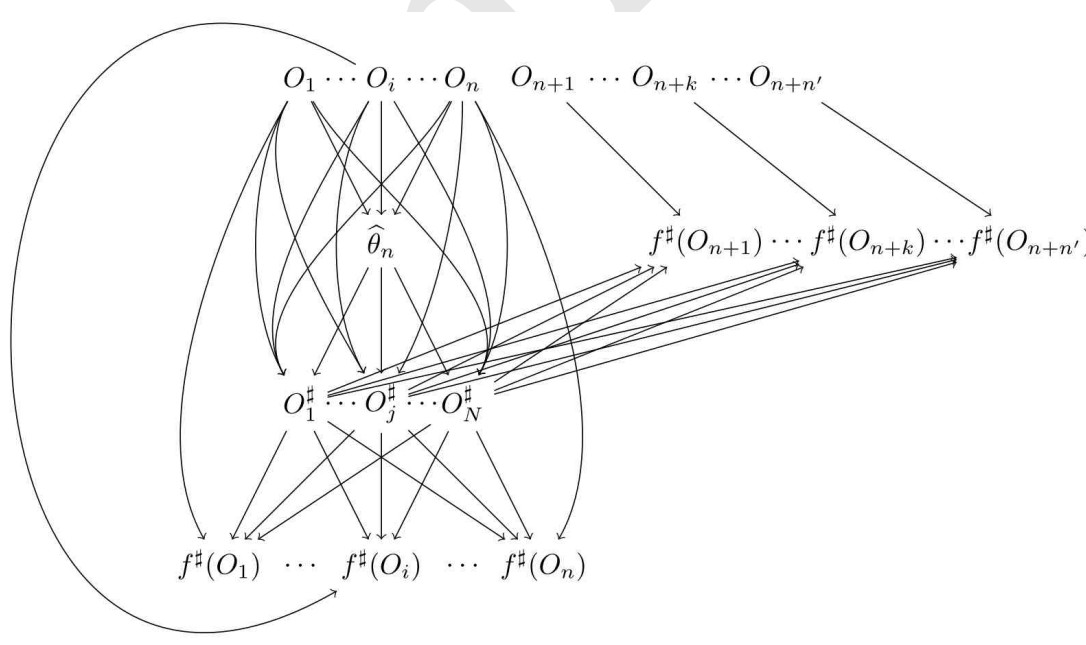

Figure 4: DAG representing how the random variables produced by $\Pi$ depend on each other.

neighbor within $\{O_1^\sharp, ..., O_N^\sharp\}$ of $O_1, ..., O_{n+n'}$ are dependent pairwise. This dependency prevents the use of a Kolmogorov-Smirnov test to compare $\mu_{1:n}$ and $\mu_{(n+1):(n+n')}$.

Moreover, conditionally on $O_1^\sharp, ..., O_N^\sharp$,

- $O_1, ..., O_n$ are not independent (because, for any $1 \le i < j \le n$, $O_1^\sharp$ is a collider on the path $O_i \to O_1^\sharp \leftarrow O_j$);
- $f^\sharp(O_1), ..., f^\sharp(O_{n+n'})$ are independent (because, for any $1 \le i < j \le n + n'$, all paths leading from $f^\sharp(O_i)$ to $f^\sharp(O_j)$ are blocked);
- the distances to the nearest neighbor within $\{O_1^\sharp, ..., O_N^\sharp\}$ of $O_{n+1}, ..., O_{n+n'}$ are mutually independent.

Therefore, conditionally on $O_1^\sharp, ..., O_N^\sharp$ and $\mu_{1:n}$, we can use t-tests to compare the three first moments of $\mu_{(n+1):(n+n')}$ to those of $\mu_{1:n}$. By the central limit theorem and Slutsky's lemma (van der Vaart 1998, Example 2.1 and Lemma 2.8), the tests are asymptotically valid as $n'$ goes to infinity.

The next chunk of code retrieves the $p$-values of the three tests using all 1000 synthetic observations.

```
# A tibble: 1 x 3
  `1st_moment_test` `2nd_moment_test` `3rd_moment_test`
            <dbl>           <dbl>           <dbl>
1         0.00706          0.0178           0.142
```

The $p$-values from the first two tests are small, but not strikingly so, especially when accounting for multiple testing. This indicates only moderate evidence of a discrepancy.

It is tempting to investigate what happens when only 100 synthetic observations are used.

```
# A tibble: 1 x 3
  `1st_moment_test` `2nd_moment_test` `3rd_moment_test`
            <dbl>           <dbl>           <dbl>
1          0.156          0.0865          0.0282
```

This time, only the $p$-value from the third tests is small, but not markedly so when accounting for multiple testing. The evidence of a discrepancy is significantly weaker when 100 synthetic observations are used compared to 1000. This highlights that distinguishing $N$ synthetic observations from genuine observations becomes increasingly difficult as $N$ decreases.

### 6.3.2 Criterion 2

The next chunk of code implements the second criterion, in its visual form.

Firstly, inspecting the first row of Figure 5 suggests that the marginal laws of $W_1, W_2, W_3$ under the synthetic law do not align very well with their counterparts under $P$, although the locations and ranges of the true marginal laws are reasonably well approximated. The restriction to $\mathbb{R}_+$ of the marginal law of $W_1$ under the synthetic law is very similar to its counterpart under $P$, but its restriction to $\mathbb{R}_-$ is too thin-tailed. As for the marginal laws of $W_2, W_3$ under the synthetic law, they are too thin-tailed compared to their counterparts under $P$.

Secondly, inspecting the second row of Figure 5 reveals that the marginal laws of $V_1, V_2$ under the synthetic law align perfectly ($V_1$) and reasonably well ($V_2$) with their counterparts under $P$. However, the marginal law of $A$ under the synthetic law assigns more weight to the event $[A = 1]$ than its counterpart under $P$.

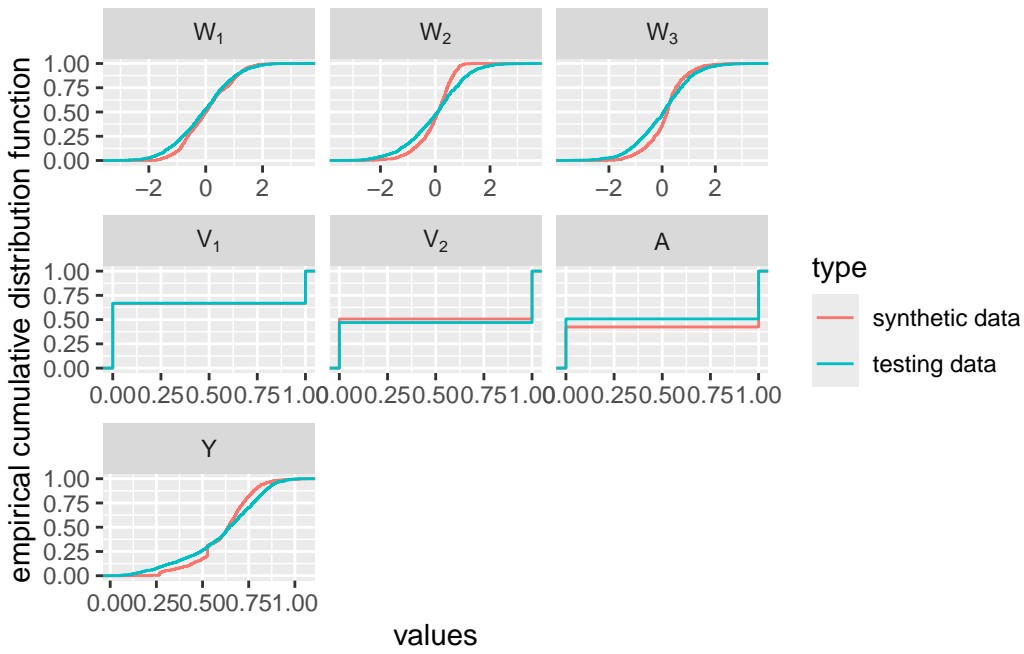

Figure 5: Empirical c.d.f. of each covariate based on either the synthetic or the testing data sets.

Lastly, inspecting the third row of Figure 5 reveals that the marginal law of $Y$ under the synthetic law does not align very well with its counterpart under $P$. While the location and range of the true marginal law are reasonably well approximated, the overall shape of the true density is not faithfully reproduced.

The next chunk of code implements the version of the second criterion based on hypotheses testing. Conditionally on the training data set, the testing procedures are valid because (i) the synthetic and testing data sets are independent, (ii) the testing data are drawn independently from $P$, (iii) the synthetic data are drawn independently from $\text{law}(\text{Gen}_{\hat{\theta}_n}(Z))$.

We first address the continuous covariates ($W_1$, $W_2$, $W_3$ and $Y$) and then the binary covariates ($V_1$, $V_2$ and $A$). For the former, we use Kolmogorov-Smirnov tests. For the latter, we use exact Fisher tests.

```
# A tibble: 4 x 2
  what     p.val
  <fct>    <dbl>
1 W[1]  6.06e- 5
2 W[2]  2.43e-14
3 W[3]  6.14e-10
4 Y     2.55e- 7

# A tibble: 3 x 2
# Groups:   what [3]
  what     p.val
  <fct>    <dbl>
1 V[1]  0.924
2 V[2]  0.117
3 A     0.000235
```

Most $p$-values are very small, supporting the conclusions drawn from inspecting Figure 5. Unlike

the marginal laws of $V_1, V_2$, the marginal laws of $W_1, W_2, W_3, A, Y$ are not well approximated, as the tests detect discrepancies when both the synthetic and testing data sets contain 1000 data points.

Naturally, one might wonder whether this result still holds when comparing smaller synthetic and testing data sets. The following chunk of code reproduces the same statistical analysis as before, but now using two samples of 100 data points each.

```
# A tibble: 4 x 2
  what   p.val
  <fct> <dbl>
1 W[1]   0.386
2 W[2]   0.678
3 W[3]   0.0590
4 Y      0.921

# A tibble: 3 x 2
# Groups:   what [3]
  what   p.val
  <fct> <dbl>
1 V[1]   0.0542
2 V[2]   0.706
3 A      1
```

This time, the *p*-values are large, indicating that the tests cannot detect discrepancies when the synthetic and testing data sets contain only 100 data points. Surprisingly, the same conclusion holds when comparing a synthetic data set of 100 data points with a testing data set of 1000 data points, as demonstrated in the next chunk of code.

```
# A tibble: 4 x 2
  what   p.val
  <fct> <dbl>
1 W[1]   0.718
2 W[2]   0.849
3 W[3]   0.0466
4 Y      0.677

# A tibble: 3 x 2
# Groups:   what [3]
  what  p.val
  <fct> <dbl>
1 V[1]   0.568
2 V[2]   0.592
3 A      0.601
```

In conclusion, while a large synthetic data set can be shown to differ in law from a large testing data set, a smaller synthetic data set does not exhibit noticeable differences in marginal laws when compared to either a small or a large testing data set.

### 6.3.3   Criterion 3

The next chunk of code builds a super learning algorithm to estimate either the conditional probability that $A = 1$ given $(W, V)$ or the conditional mean of $Y$ given $(A, W, V)$ by aggregating 5 base learners. We use the SuperLearner package in R. Specifically, the 5 base learners estimate the above conditional means by a constant (SL.mean), or based on generalized linear models (SL.glm and

SL.glm.interaction), or by a random forest (SL.ranger), or based on a single-hidden-layer neural network (SL.nnet).

We train the super learning algorithm three times: once on each of two distinct halves of the training data set, and once on half of the synthetic data set. This results in three estimators of the conditional probability that $A = 1$ given $(W, V)$ and three estimators of the conditional mean of $Y$ given $(A, W, V)$. The next chunk of code prepares the three training data sets.

```
# A tibble: 3 x 3
  type                data             testing
  <chr>               <list>           <list>
1 using training data a <tibble [500 x 7]> <tibble [1,000 x 7]>
2 using training data b <tibble [500 x 7]> <tibble [1,000 x 7]>
3 using synthetic data  <tibble [500 x 7]> <tibble [1,000 x 7]>
```

The following chunk of code trains the super learning algorithm and evaluates the six resulting estimators on the testing data points. To compare the estimators, we use scatter plots. Specifically, denoting by $\widehat{pr}_1, \widehat{pr}_2, \widehat{pr}_3$ the estimators of the conditional probability that $A = 1$ given $(W, V)$ obtained by training the super learning algorithm on each of the two distinct halves of the training data set ($\widehat{pr}_1$ and $\widehat{pr}_2$), and on half of the synthetic data set ($\widehat{pr}_3$), we plot in the left-hand side panel $\{(\widehat{pr}_1(W_{n+i}, V_{n+i}), \widehat{pr}_2(W_{n+i}, V_{n+i})) : 1 \le i \le n\}$ (in red) and $\{(\widehat{pr}_1(W_{n+i}, V_{n+i}), \widehat{pr}_3(W_{n+i}, V_{n+i})) : 1 \le i \le n\}$ (in blue).

Therein, the spread of the red scatter plot along the $y = x$ line in the $xy$-plane is an evidence of the inherent and irreducible randomness that one faces when one learns the conditional probability that $A = 1$ given $(W, V)$. By comparison, the blue scatter plot is more widely spread around the line, revealing a measure of discrepancy between the training and synthetic data.

The right-hand side panel is obtained analogously. The red scatter plot is more concentrated around the $y = x$ line than its counterpart in the left-hand side panel. The blue scatter plot is more widely spread than the red one, which again reveals a measure of discrepancy between the training and synthetic data. In summary, we consider that the red and blue scatter plots do not strongly differ in their bulks. However, it seems that the blue scatter plots feature more outliers than their red counterparts, revealing that the estimators may be quite different in some parts of the space of covariates.

### 6.3.4 Summary

We implement three criteria to evaluate the synthetic observations. The first criterion compared empirical distributions of distances between genuine observations, both involved and not involved in the generator's construction, and synthetic observations, detecting minor over-replication in the synthetic data set. The second criterion assessed marginal distributions of individual features, revealing discrepancies, particularly in continuous variables, which often exhibited overly thin tails. The third criterion compared predictions from an algorithm trained on synthetic versus genuine observations, showing good replication for predicting $Y$ given $(A, W, V)$ but less so for predicting $A$ given $(W, V)$. Overall, while the synthetic observations show some discrepancies from the genuine ones, these differences are not overly substantial. Moreover, detecting significant differences becomes much harder with smaller synthetic datasets (100 versus 1000 synthetic observations).

## 7  Illustration on real data

In this section, we extend the analysis conducted in the previous section to real data. We use a subset of the International Warfarin Pharmacogenetics Consortium IWPC data set (The International Warfarin

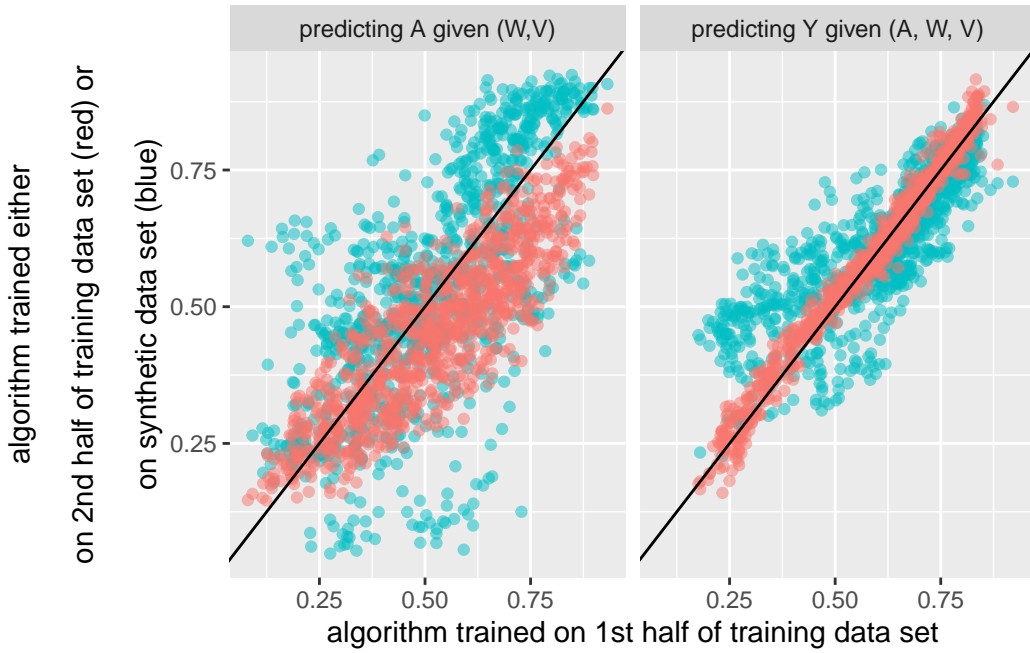

Figure 6: Comparing predicted conditional probabilities that $A = 1$ given $(W, V)$ (left) or predicted conditional means of $Y$ given $(A, W, V)$ (right) when a super learning algorithm is trained twice on two distinct halves of the training data set (red points) or on the first half of the training data set, $x$-axis, and on half of the synthetic data set, $y$-axis (blue points).

Pharmacogenetics Consortium 2009). Warfarin therapy is a commonly prescribed anticoagulant employed to treat thrombosis and thromboembolism.

## 7.1 The International Warfarin Pharmacogenetics Consortium data set

In order to limit the number of incomplete observations, we keep only the following variables:

- height, in centimeters;
- weight, in kilograms;
- indicator of whether or not VKORC1 consensus (obtained from genotype data) is "A/A";
- indicator of whether or not CYP2C9 consensus (obtained from genotype data) is "*1/*1";
- indicator of whether or not ethnicity is Asian;
- indicator of whether or not therapeutic dose of Warfarin is greater than or equal to 21 mg;
- international normalized ratio on reported therapeutic dose of Warfarin (INR, a measure of blood clotting function).

The original database includes 3193 patients with complete observations for these variables. We refer to the table below for a brief description of the data.

Let us load the data set into `Python`.

It is convenient to rescale the continuous variables.

We finally define the training and testing data sets.

```
The three first observations in 'train':
  V_1  V_2  V_3  W_1   W_2   A    Y
 [[1.   0.   0.   0.688 0.515 1.    0.121]
```

Table 1

| Variables | | Descriptive statistics | | | | |
|---|---|---|---|---|---|---|
| | | n (%) or Min | Median | Mean | Max | SD |
| V1 | VKORC1 consensus is A/A | 1,150 (36%) | | | | |
| V2 | CYP2C9 consensus is *1/*1 | 2,467 (77%) | | | | |
| V3 | Ethnicity is Asian | 1,087 (34%) | | | | |
| W1 | Height (cm) | 125 | 168 | 168 | 202 | 10.9 |
| W2 | Weight (kg) | 30 | 73.0 | 76.9 | 238 | 22.0 |
| A | Therapeutic dose >= 21 mg per week | 2,336 (73%) | | | | |
| Y | INR | 4 | 65.3 | 74.1 | 680 | 47.1 |

```
722  [0.    0.    1.    0.584 0.13  0.    0.029]
723  [0.    0.    1.    0.402 0.169 0.    0.04 ]]
```

## 7.2 Training the VAE

By running the next chunk of code, we set the VAE's configuration.

The next chunk of code repeatedly generates and initializes a VAE then trains it.

Because running the chunk is time-consuming, we stored one trained VAE that we considered good enough. We now turn to its evaluation based on the three criteria discussed in Section 6.2 and Section 6.3.

## 7.3 Evaluating the quality of the generator

The next chunk of code defines in R the counterparts train and test of the Python objects train and test (keeping only the first 1000 observations), and synth, the collection of 1000 synthetic observations drawn from the generator associated to the VAE that we stored in Section 7.2. For later use (while implementing Criterion 1) we add a dummy column named Z.

### 7.3.1 Criterion 1

The next chunk of code implements the first criterion.

The two empirical c.d.f. shown in Figure 7 are not as similar as those in Figure 3. The next chunk of code implements the t-tests comparing the three first moments of $\mu_{(n+1):(n+n')}$ to those of $\mu_{1:n}$.

```
739  # A tibble: 1 x 3
740    `1st_moment_test` `2nd_moment_test` `3rd_moment_test`
741             <dbl>             <dbl>             <dbl>
742  1       7.59e-13          2.47e-18          8.18e-26
```

The numerical evidence of discrepancy is compelling. But is it still as compelling when only 100 synthetic observations are used? The next chunk of code addresses this question.

```
745  # A tibble: 1 x 3
746    `1st_moment_test` `2nd_moment_test` `3rd_moment_test`
747             <dbl>             <dbl>             <dbl>
748  1        0.0330            0.0135           0.00313
```

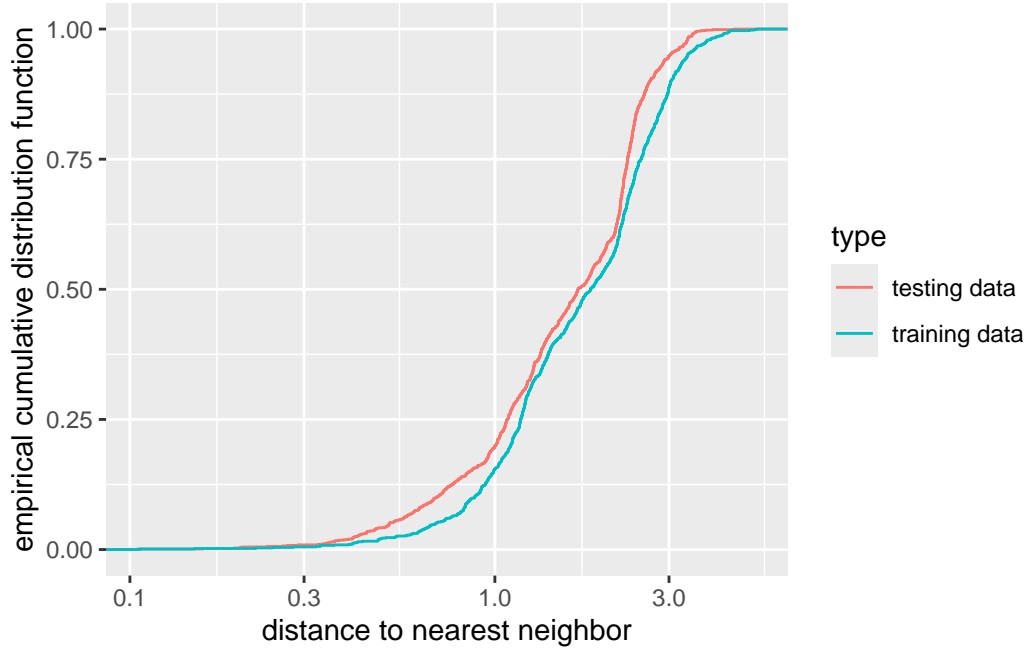

Figure 7: Empirical c.d.f. of the distance to the nearest neighbor within the synthetic observations of the training and of the testing IWPC data points (logarithmic scale). The two c.d.f. are quite close.

The strength of evidence has dropped considerably, reflected by the larger $p$-values compared to earlier results. As in Section 6.3, distinguishing $N$ synthetic observations from genuine observations becomes more challenging when $N = 100$ compared to $N = 1000$.

### 7.3.2 Criterion 2

The next chunk of code implements the second criterion, in its visual form.

Figure 8 suggests that except for $V_2$, $A$ and, to a lesser extent, $V_1$, the marginal laws under the synthetic law do not align well with their counterparts under $P$. This is confirmed by the following (Kolmogorov-Smirnov or exact Fisher) hypotheses tests:

```
# A tibble: 3 x 2
  what      p.val
  <fct>     <dbl>
1 W[1]  1.93e- 25
2 W[2]  5.53e-180
3 Y     4.37e- 32

# A tibble: 4 x 2
# Groups:   what [4]
  what      p.val
  <fct>     <dbl>
1 V[1]  1.08e- 3
2 V[2]  5.99e- 1
3 V[3]  2.23e-22
4 A     3.59e- 1
```

One might again question whether this result persists when comparing smaller synthetic and testing datasets. The next chunk of code replicates the previous statistical analysis, this time using two

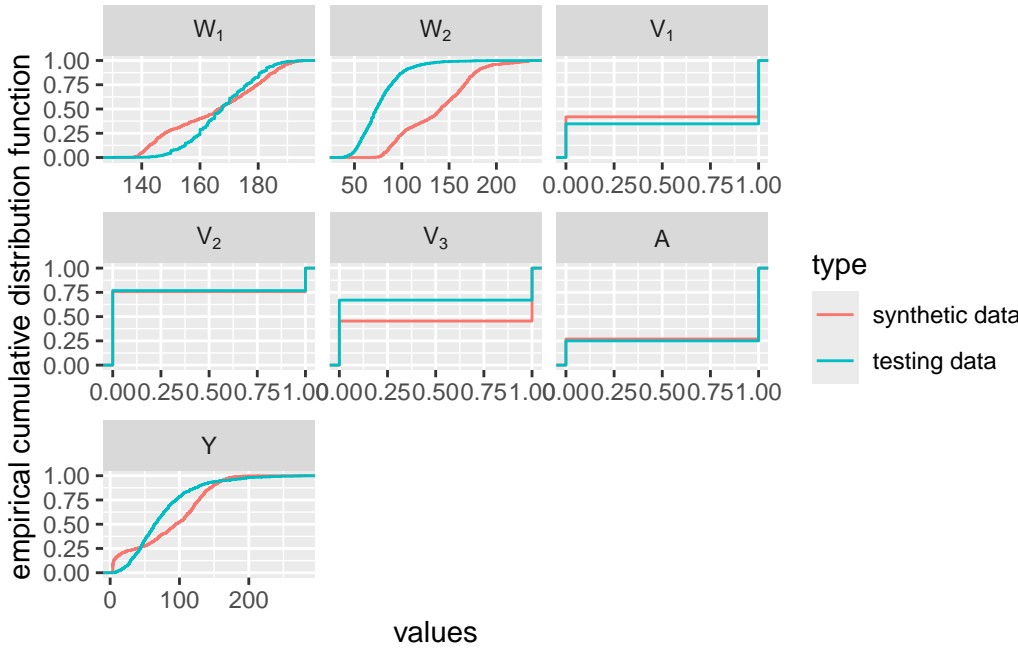

Figure 8: Empirical c.d.f. of each covariate based on either the synthetic observations or the testing IPWC data set.

samples of 100 data points each.

```
# A tibble: 3 x 2
  what    p.val
  <fct>   <dbl>
1 W[1]  0.373
2 W[2]  0.00184
3 Y     0.0590
```

```
# A tibble: 4 x 2
# Groups:   what [4]
  what  p.val
  <fct> <dbl>
1 V[1]  1
2 V[2]  0.596
3 V[3]  1
4 A     1
```

This time, except for $W_2$, the $p$-values are large, indicating that the tests cannot detect discrepancies when the synthetic and testing data sets each contain only 100 data points. As observed in Section 6.3, a similar conclusion holds when comparing a synthetic dataset of 100 data points with a testing dataset of 1000 data points, with $Y$ now also associated with a small $p$-value. This is demonstrated by the next chunk of code.

```
# A tibble: 3 x 2
  what       p.val
  <fct>      <dbl>
1 W[1]  0.129
2 W[2]  0.000000179
```

```
798  3 Y      0.00730
799  # A tibble: 4 x 2
800  # Groups:   what [4]
801    what   p.val
802    <fct> <dbl>
803  1 V[1]   1
804  2 V[2]   1
805  3 V[3]   0.0830
806  4 A      1
```

In conclusion, although a large synthetic dataset can be shown to differ in distribution from a large testing dataset, a smaller synthetic dataset does not display clear differences in marginal distributions (apart from $W_3$ and potentially $Y$) when compared to a small testing dataset.

### 7.3.3 Criterion 3

The next chunk of code builds a super learning algorithm to estimate either the conditional probability that $A = 1$ given $(W, V)$ or the conditional mean of $Y$ given $(A, W, V)$ by aggregating the same 5 base learners as in Section 6.3.

We train the super learning algorithm three times: once on each of two distinct halves of the training data set, and once on half of the synthetic data set. This results in three estimators of the conditional probability that $A = 1$ given $(W, V)$ and three estimators of the conditional mean of $Y$ given $(A, W, V)$. The next chunk of code prepares the three training data sets.

```
818  # A tibble: 3 x 3
819    type                 data                testing
820    <chr>                <list>              <list>
821  1 using training data a <tibble [500 x 7]> <tibble [1,000 x 7]>
822  2 using training data b <tibble [500 x 7]> <tibble [1,000 x 7]>
823  3 using synthetic data  <tibble [500 x 7]> <tibble [1,000 x 7]>
```

The following chunk of code trains the super learning algorithm and evaluates the six resulting estimators on the testing data points. To compare the estimators, we use scatter plots in the same manner as in Section 6.3.

Therein, the spread and asymmetry of the red scatter plot along the $y = x$ line in the $xy$-plane are evidences of how difficult it is to estimate the conditional probability that $A = 1$ given $(W, V)$. To ease comparisons, we also superimpose the regression lines obtained by fitting two separate linear models on the blue and red data points. By comparison, the blue scatter plot is less widely spread than the red one, around the blue line which deviates more from the $y = x$ line than the red one.

The right-hand side panel is obtained analogously. The red scatter plot is more concentrated around the $y = x$ line than its counterpart in the left-hand side panel. This indicates that it is less difficult to estimate the conditional mean of $Y$ given $(A, W, V)$ than the probability that $A = 1$ given $(W, V)$. The blue scatter plot is more widely spread than the red one, which again reveals a measure of discrepancy between the training and synthetic data. This is counterbalanced by the fact that the blue regression line almost coincides with the $y = x$ line, whereas the red one deviates from it.

### 7.4 Summary

We implemented the same three criteria as in Section 6.3. Overall, the synthetic observations showed more substantial discrepancies from the IWPC (genuine) ones compared to the analysis on simulated

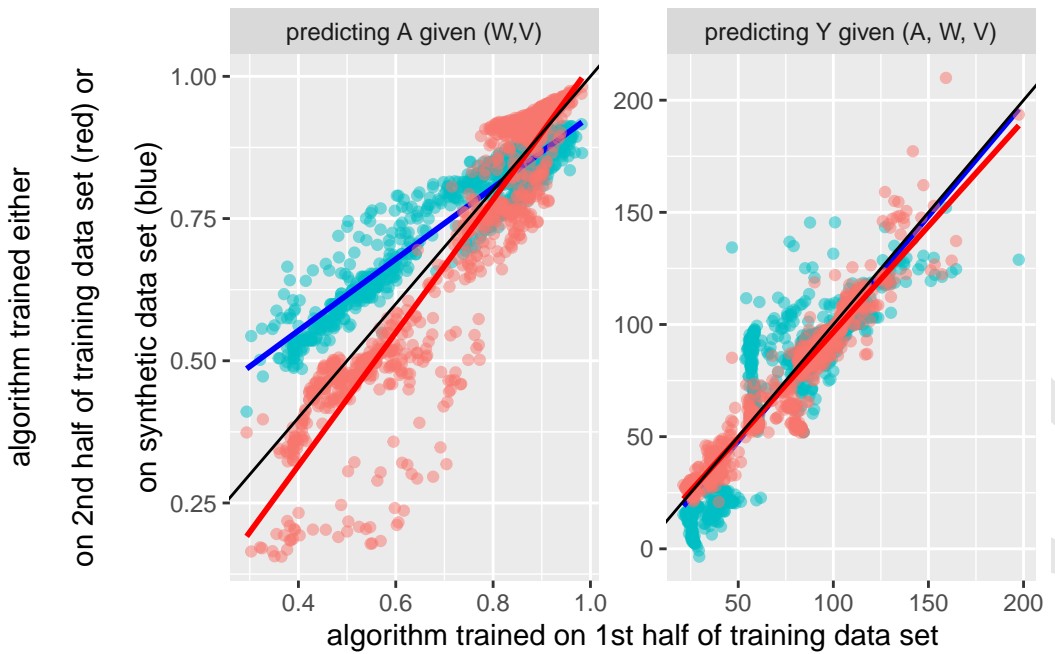

Figure 9: Comparing predicted conditional probabilities that $A = 1$ given $(W, V)$ (left) or predicted conditional means of $Y$ given $(A, W, V)$ (right) when a super learning algorithm is trained twice on two distinct halves of the training IWPC data set (red points) or on the first half of the training IWPC data set, $x$-axis, and on half of the synthetic data set, $y$-axis (blue points).

data. Furthermore, detecting significant differences remained challenging with smaller synthetic datasets (100 versus 1000 synthetic observations), but the gaps were more evident in the real-data context.

# 8  Conclusion

This final section contextualizes our study by reviewing related works, discussing the challenges and limitations encountered, and offering a closing reflection on the broader implications of our approach and findings.

## 8.1  Related works

Before the advent of neural networks, synthetic tabular data were typically generated by modeling the joint law of genuine tabular data and sampling from it. The parametric models involved canonical distributions and were often restricted to low-dimensional settings, due to computational limitations and the challenges of effectively encoding large, parameterized classes of functions.

With the emergence of neural networks, numerous studies have focused on generating synthetic data across diverse fields, including image generation (Yi, Walia, and Babyn 2019), video synthesis (Vondrick, Pirsiavash, and Torralba 2016), natural language processing (Lee 2018), and healthcare (Che et al. 2017), (Choi et al. 2017), (Baowaly et al. 2018), (Lee 2018), (Yi, Walia, and Babyn 2019). Most of these studies employed Generative Adversarial Networks (GANs, Goodfellow et al. (2014)), for instance (Creswell et al. 2018), (Gui et al. 2023), (Aggarwal, Mittal, and Battineni 2021), (Figueira and Vaz 2022), though VAEs and their extensions were also used.

For instance, Xu et al. (2019) proposed the Conditional Tabular GAN (CTGAN) to address challenges

specific to tabular data, such as the mix of discrete and continuous variables, multiple modes in continuous variables, and imbalanced discrete variables. Their approach included mode-specific normalization, architectural modifications, a conditional generator, and training-by-sampling to improve performance. Additionally, they introduced the Tabular Variational Auto-Encoder (TVAE) for mixed-type tabular data generation.

Inspired by a randomized controlled trial (RCT) in the treatment of Human Immunodeficiency Virus (HIV), Petrakos, Moodie, and Savy (2025) recently conducted an empirical comparison of several strategies and two data generation techniques aimed at generating synthetic tabular RCT data while preserving the underlying multivariate data distribution. One of these techniques was based on the aforementioned CTGAN, and the other on a more traditional statistical method. Their findings indicate that the most effective approach for generating synthetic RCT data involves a sequential generation process. This process begins with an R-vine copula model to generate baseline variables, followed by a simple random treatment allocation to simulate the RCT environment, and concludes with regression models for post-treatment variables, such as the trial outcome.

In a causal framework, Kocaoglu et al. (2017) proposed an adversarial training procedure to learn a causal implicit generative model for a given causal graph. They demonstrated that when the generator's structure aligns with the causal graph, GANs can effectively train causal implicit generative models. Their approach involved a two-stage procedure: first, they trained a causal implicit generative model over binary labels using a Wasserstein GAN (WGAN, Arjovsky, Chintala, and Bottou (2017), Gulrajani et al. (2017)) consistent with the causal graph as the generator. Next, they introduced a novel conditional GAN architecture, called CausalGAN, which incorporates an anti-labeler network alongside a labeler network in its loss function. They showed that this architecture enables sampling from the correct conditional and interventional distributions.

Works more similar to ours include the following. Athey et al. (2024) proposed using GANs to generate synthetic data that mimic genuine data, aimed at assessing the performance of statistical methods. To illustrate their approach, they employed WGANs to generate the covariates $(V, W)$ conditional on the treatment $A$, followed by the outcome variable $Y$ conditional on $A$. The resulting synthetic data were used to estimate average treatment effects.

Neal, Huang, and Raghupathi (2021) presented RealCause, an alternative approach to simulate realistic data using neural networks. Unlike our approach, they first sampled $(V, W)$ directly from the genuine data and then generated samples for $A$ (conditionally on $(W, V)$) and $Y$ (conditionally on $(A, W, V)$).

Parikh et al. (2022) introduced Credence, a deep generative model-based framework for evaluating causal inference methods. A distinctive feature of their approach is its ability to specify ground truth for both the form and magnitude of causal effects and confounding bias as functions of covariates. Like us, they used a VAE but modeled the joint law of $(V, W, A, Y)$ by decomposing it into the two conditional laws of $Y$ given $(A, W, V)$ and of $(V, W)$ given $A$, along with the marginal law of $A$. However, their decomposition of the likelihood differs from ours.

Naturally, researchers have also focused on evaluating the quality of synthetic data. For instance, Algorithm ?? proposed three metrics ($\alpha$-precision, $\beta$-recall, authenticity) to assess the fidelity, diversity, and generalization performance of data generators. In their work, each sample is individually classified as either high-quality or low-quality.

To conclude, Lu et al. (2024) provided a comprehensive systematic review of studies utilizing machine learning models to generate synthetic data, offering a valuable synthesis of the field's progress and challenges.

## 8.2 Challenges and limitations

The results of our study, while informative, are somewhat disappointing. Increasing the quantity of genuine data substantially did not improve the simulator's performance in this context. This is in stark contrast to fields like image generation, where the abundance of inherent regularities in visual patterns enables models to learn effectively from larger datasets. In our case, the limited improvement may stem from a lack of rich regularities in the genuine data, which constrains the simulator's ability to capture meaningful structures.

Another challenge lies in the question of sharing the simulator. While it would be appealing to make the simulator widely available, doing so raises concerns about the genuine data required to run the code. This dependency could potentially compromise the privacy or utility of the original dataset, creating additional barriers to adoption.

It is also worth noting that we deliberately neutralized in this article the VAE's repeated training from random initializations due to the high computational time required. This is telling to the extent that the computational cost underscores a practical limitation of the approach: the trade-off between feasibility and the potential benefits of repeated and extended training cycles, which might otherwise improve the simulator's performance.

Looking ahead, addressing some of these limitations requires practical and theoretical advances. For instance, future efforts could focus on effectively handling missing data (NA values) within the simulation framework. Additionally, establishing general design principles for simulator architectures could improve their robustness and adaptability across a variety of datasets and applications.

## 8.3 Closing reflection

As we reflect on the limitations and implications of simulators, it is worth revisiting the paragraph in Section 1.4 where we state that parametric simulators "cannot convincingly replicate the multifaceted interactions and variability inherent in 'nature' ". The lexical field surrounding "nature" itself warrants reflection.

Historically, the notion of "nature" has evolved significantly. In ancient Greece, philosophers used the term "physis", nowadays often translated as "nature", to explore the inherent essence or intrinsic qualities of things. "Natura", the Roman adaptation, extended these ideas, while medieval thought integrated nature into theological frameworks, portraying it as divine creation.

Often regarded as a figure of the late Renaissance and an early architect of the Scientific Revolution, Bacon emphasized in 1620 the idea of conquering nature, viewing it as an object to be studied, understood, and controlled – "for nature is only to be commanded by obeying her" (Bacon 1854). During the Enlightenment, the concept of "nature" further shifted, increasingly separating it from humanity and framing it as an object of scientific study and exploitation. These developments have frequently served as a conceptual tool to justify humanity's dominion and looting of the non-human world.

In this context, referring to "nature" as something simulations seek to imitate is a testament to the evolving notion of "nature," now encompassing phenomena like human health. This shift should be questioned, especially if it follows the Enlightenment logic of treating "nature" as an object to be understood, controlled, and exploited. Applying such a framework to humans risks reducing individuals to abstract data points or exploitable systems, ignoring their intrinsic complexity and moral agency.

Recognizing these risks invites us to critically examine not only the limitations of simulators but also their ethical and philosophical implications. Among these are the challenges posed by a lack of

fair representability in the data used to train algorithms, which can perpetuate existing inequities or create new ones. Fairness becomes a central issue, particularly when simulations influence decisions that affect diverse populations, as the underlying models may not account for all relevant perspectives or experiences. Furthermore, the use of advanced simulations can contribute to elitism, as access to the expertise and computational resources needed to develop and deploy such systems is far from being universal, with numerous countries facing more urgent challenges. Finally, the environmental and financial cost of training complex algorithms, particularly those based on generative AI, raises questions about sustainability and the trade-offs between progress and resource consumption.

Fiction has always provided a space to explore hypothetical scenarios that might be impractical, impossible, or even unethical in reality. Utopian and dystopian literature, for example, simulates alternative societies to test ideas about governance, morality, and human behavior. Similarly, speculative fiction pushes boundaries by imagining futures shaped by scientific and technological advancements. In doing so, fiction serves as a conceptual laboratory, allowing its creators and audiences to investigate possibilities and their consequences. This creative exploration, which has long shaped human understanding, should continue to inform and inspire the design and purpose of computer simulations.

# 9   Acknowledgements

A number of open-source libraries made this work possible. In `Python`, we relied on the packages `numpy` (Harris et al. 2020), `pandas` (McKinney 2010), `random` (Van Rossum 2020), `sklearn` (Pedregosa et al. 2011) and `tensorflow` (Abadi et al. 2015). In `R`, we used the packages `gtsummary` (Sjoberg et al. 2021), `kknn` (Schliep and Hechenbichler 2016), `SuperLearner` (Polley et al. 2024) and `tidyverse` (Wickham et al. 2019). We are grateful to the developers and maintainers of these packages for their contributions to the research community.

The authors warmly thank Isabelle Drouet (Sorbonne Université) and Alexander Reisach (Université Paris Cité) for their valuable feedback and insightful comments on this project.

Abadi, Martín, Ashish Agarwal, Paul Barham, Eugene Brevdo, Zhifeng Chen, Craig Citro, Greg S. Corrado, et al. 2015. "TensorFlow: Large-Scale Machine Learning on Heterogeneous Systems." https://www.tensorflow.org/.

Aggarwal, Alankrita, Mamta Mittal, and Gopi Battineni. 2021. "Generative Adversarial Network: An Overview of Theory and Applications." *International Journal of Information Management Data Insights* 1 (1): 100004. https://doi.org/https://doi.org/10.1016/j.jjimei.2020.100004.

Aigner, Martin, and Günter M. Ziegler. 2018. *Proofs from THE BOOK*. Springer.

Aristote. 2006. *Poétique*. Paris: Édition Mille et une nuits.

Arjovsky, Martin, Soumith Chintala, and Léon Bottou. 2017. "Wasserstein Generative Adversarial Networks." In *Proceedings of the 34th International Conference on Machine Learning*, edited by Doina Precup and Yee Whye Teh, 70:214–23. Proceedings of Machine Learning Research. PMLR.

Arora, Sanjeev, Nadav Cohen, and Elad Hazan. 2018. "On the Optimization of Deep Networks: Implicit Acceleration by Overparameterization." In *International Conference on Machine Learning*, 244–53. PMLR.

Athey, Susan, Guido W. Imbens, Jonas Metzger, and Evan Munro. 2024. "Using Wasserstein Generative Adversarial Networks for the Design of Monte Carlo Simulations." *Journal of Econometrics* 240 (2): 105076. https://doi.org/https://doi.org/10.1016/j.jeconom.2020.09.013.

Bacon, Francis. 1854. *Novum Organum 1620*. Vol. 3. Philadelphia: Parry & MacMillan. https://history.hanover.edu/texts/bacon/novorg.html.

Baowaly, Mrinal Kanti, Chia-Ching Lin, Chao-Lin Liu, and Kuan-Ta Chen. 2018. "Synthesizing Electronic Health Records Using Improved Generative Adversarial Networks." *Journal of the*

American Medical Informatics Association 26: 228–41.

Barberousse, Anouk, and Pascal Ludwig. 2000. "Les Modèles Comme Fiction." *Philosophie* 68: 16–43.

Carruthers, Peter. 2002. "Human Creativity: Its Cognitive Basis, Its Evolution, and Its Connections with Childhood Pretence." *British Journal for the Philosophy of Science* 53: 225–49.

Che, Zhengping, Yu Cheng, Shuangfei Zhai, Zhaonan Sun, and Yan Liu. 2017. "Boosting Deep Learning Risk Prediction with Generative Adversarial Networks for Electronic Health Records." In *2017 IEEE International Conference on Data Mining (ICDM)*, 787–92. https://doi.org/10.1109/ICDM.2017.93.

Choi, Edward, Siddharth Biswal, Bradley Malin, Jon Duke, Walter F. Stewart, and Jimeng Sun. 2017. "Generating Multi-Label Discrete Patient Records Using Generative Adversarial Networks." In *Proceedings of the 2nd Machine Learning for Healthcare Conference*, edited by Finale Doshi-Velez, Jim Fackler, David Kale, Rajesh Ranganath, Byron Wallace, and Jenna Wiens, 68:286–305. Proceedings of Machine Learning Research. PMLR. https://proceedings.mlr.press/v68/choi17a.html.

Choromanska, Anna, Mikael Henaff, Michael Mathieu, Gérard Ben Arous, and Yann LeCun. 2015. "The Loss Surfaces of Multilayer Networks." In *Artificial Intelligence and Statistics*, 192–204. PMLR.

Creswell, Antonia, Tom White, Vincent Dumoulin, Kai Arulkumaran, Biswa Sengupta, and Anil A. Bharath. 2018. "Generative Adversarial Networks: An Overview." *IEEE Signal Processing Magazine* 35 (1): 53–65. https://doi.org/10.1109/MSP.2017.2765202.

de Saint-Exupéry, Antoine. 1943. *Le Petit Prince.* New-York: Reynal & Hitchcock.

Figueira, Alvaro, and Bruno Vaz. 2022. "Survey on Synthetic Data Generation, Evaluation Methods and GANs." *Mathematics* 10 (15). https://doi.org/10.3390/math10152733.

Glorot, Xavier, and Yoshua Bengio. 2010. "Understanding the Difficulty of Training Deep Feedforward Neural Networks." In *Proceedings of the Thirteenth International Conference on Artificial Intelligence and Statistics*, edited by Yee Whye Teh and Mike Titterington, 9:249–56. Proceedings of Machine Learning Research. Chia Laguna Resort, Sardinia, Italy: PMLR. https://proceedings.mlr.press/v9/glorot10a.html.

Goodfellow, Ian, Jean Pouget-Abadie, Mehdi Mirza, Bing Xu, David Warde-Farley, Sherjil Ozair, Aaron Courville, and Yoshua Bengio. 2014. "Generative Adversarial Nets." In *Advances in Neural Information Processing Systems*, edited by Z. Ghahramani, M. Welling, C. Cortes, N. Lawrence, and K. Q. Weinberger. Vol. 27. Curran Associates, Inc.

Gui, Jie, Zhenan Sun, Yonggang Wen, Dacheng Tao, and Jieping Ye. 2023. "A Review on Generative Adversarial Networks: Algorithms, Theory, and Applications." *IEEE Transactions on Knowledge and Data Engineering* 35 (4): 3313–32. https://doi.org/10.1109/TKDE.2021.3130191.

Gulrajani, Ishaan, Faruk Ahmed, Martin Arjovsky, Vincent Dumoulin, and Aaron C Courville. 2017. "Improved Training of Wasserstein GANs." In *Advances in Neural Information Processing Systems*, edited by I. Guyon, U. Von Luxburg, S. Bengio, H. Wallach, R. Fergus, S. Vishwanathan, and R. Garnett. Vol. 30. Curran Associates, Inc.

Harris, Charles R., K. Jarrod Millman, Stéfan J.van der Walt, Ralf Gommers, Pauli Virtanen, David-Cournapeau, Eric Wieser, et al. 2020. "Array Programming with NumPy." *Nature* 585 (7825): 357–62. https://doi.org/10.1038/s41586-020-2649-2.

Homère. 2000. *L'Odyssée.* Paris: La Découverte.

Kingma, Diederik P., and Max Welling. 2014. "Auto-Encoding Variational Bayes." In *2nd International Conference on Learning Representations, ICLR 2014, Banff, AB, Canada, April 14-16, 2014, Conference Track Proceedings*, edited by Yoshua Bengio and Yann LeCun.

Kocaoglu, Murat, Christopher Snyder, Alexandros G. Dimakis, and Sriram Vishwanath. 2017. "CausalGAN: Learning Causal Implicit Generative Models with Adversarial Training." *CoRR* abs/1709.02023. http://arxiv.org/abs/1709.02023.

Lauritzen, Steffen L. 1996. *Graphical Models.* Vol. 17. Oxford Statistical Science Series. The Clarendon

1046    Press, Oxford University Press, New York.

1047    Lee, Scott H. 2018. "Natural Language Generation for Electronic Health Records." *Npj Digital Medicine*
1048    1 (1). https://doi.org/10.1038/s41746-018-0070-0.

1049    Lu, Yingzhou, Minjie Shen, Huazheng Wang, Xiao Wang, Capucine van Rechem, Tianfan Fu, and
1050    Wenqi Wei. 2024. "Machine Learning for Synthetic Data Generation: A Review." https://arxiv.
1051    org/abs/2302.04062.

1052    McKinney, Wes. 2010. "Data Structures for Statistical Computing in Python." In *Proceedings of*
1053    *the 9th Python in Science Conference*, edited by Stéfan van der Walt and Jarrod Millman, 56–61.
1054    https://doi.org/ 10.25080/Majora-92bf1922-00a .

1055    Metropolis, Nicholas, and Stanislaw Ulam. 1949. "The Monte Carlo Method." *Journal of the American*
1056    *Statistical Association* 44 (247): 335–41.

1057    Morris, T. P., I. R. White, and M. J. Crowther. 2019. "Using Simulation Studies to Evaluate Statistical
1058    Methods." *Statistics in Medicine* 38. https://doi.org/DOI: 10.1002/sim.8086.

1059    Neal, Brady, Chin-Wei Huang, and Sunand Raghupathi. 2021. "RealCause: Realistic Causal Inference
1060    Benchmarking." https://arxiv.org/abs/2011.15007.

1061    Parikh, Harsh, Carlos Varjao, Louise Xu, and Eric Tchetgen Tchetgen. 2022. "Validating Causal
1062    Inference Methods." In *Proceedings of the 39th International Conference on Machine Learning*,
1063    edited by Kamalika Chaudhuri, Stefanie Jegelka, Le Song, Csaba Szepesvari, Gang Niu, and Sivan
1064    Sabato, 162:17346–58. Proceedings of Machine Learning Research. PMLR.

1065    Pearl, Judea, and Dana Mackenzie. 2018. *The Book of Why: The New Science of Cause and Effect.*
1066    New-York: Basic Books.

1067    Pedregosa, Fabian, Gaël Varoquaux, Alexandre Gramfort, Vincent Michel, Bertrand Thirion, Olivier
1068    Grisel, Mathieu Blondel, et al. 2011. "Scikit-Learn: Machine Learning in Python." *Journal of*
1069    *Machine Learning Research* 12 (Oct): 2825–30.

1070    Petrakos, Niki Z., Erica E. M. Moodie, and Nicolas Savy. 2025. "A Framework for Generating Realistic
1071    Synthetic Tabular Data in a Randomized Controlled Trial Setting." https://arxiv.org/abs/2501.
1072    17719.

1073    Platon. 2002. *La République.* Paris: Flammarion.

1074    Polley, Eric, Erin LeDell, Chris Kennedy, and Mark van der Laan. 2024. *SuperLearner: Super Learner*
1075    *Prediction.* https://CRAN.R-project.org/package=SuperLearner.

1076    R Core Team. 2020. *R: A Language and Environment for Statistical Computing.* Vienna, Austria: R
1077    Foundation for Statistical Computing. https://www.R-project.org/.

1078    Raspe, Rudolf E. 1866. *Aventures Du Baron de Münchausen.* Paris: Furne, Jouvet et Cie. https:
1079    //gallica.bnf.fr/ark:/12148/bpt6k6582615r.

1080    Rezende, Danilo Jimenez, Shakir Mohamed, and Daan Wierstra. 2014. "Stochastic Backpropagation
1081    and Approximate Inference in Deep Generative Models." In *Proceedings of the 31st International*
1082    *Conference on Machine Learning*, edited by Eric P. Xing and Tony Jebara, 32:1278–86. Proceed-
1083    ings of Machine Learning Research 2. Bejing, China: PMLR. http://proceedings.mlr.press/v32/
1084    rezende14.html.

1085    Rostand, Edmond. 2005. *Cyrano de Bergerac.* Paris: E. Fasquelle. https://gallica.bnf.fr/ark:/12148/
1086    bpt6k64960772.

1087    Schliep, Klaus, and Klaus Hechenbichler. 2016. *kknn: Weighted k-Nearest Neighbors.* https://CRAN.R-
1088    project.org/package=kknn.

1089    Shelley, Mary. 1818. *Frankenstein; or, The Modern Prometheus.* London: Lackington, Hughes, Harding,
1090    Marvor & Jones.

1091    Sjoberg, Daniel D., Karissa Whiting, Michael Curry, Jessica A. Lavery, and Joseph Larmarange.
1092    2021. "Reproducible Summary Tables with the gtsummary Package." *The R Journal* 13: 570–80.
1093    https://doi.org/10.32614/RJ-2021-053.

1094    Solly, Meilan. 2023. "The Real History Behind the Archimedes Dial in 'Indiana Jones and the Dial of
1095    Destiny'." *Smithsonian Magazine.* https://www.smithsonianmag.com/history/the-real-history-

behind-archimedes-dial-in-indiana-jones-and-the-dial-of-destiny-180982435/.

The International Warfarin Pharmacogenetics Consortium. 2009. "Estimation of the Warfarin Dose with Clinical and Pharmacogenetic Data." *New England Journal of Medicine* 360 (8): 753–64. https://doi.org/10.1056/NEJMoa0809329.

Tokarczuk, Olga. 2021. *The Books of Jacob.* Melbourne: The Text Publishing Company.

van der Vaart, Aad W. 1998. *Asymptotic Statistics.* Vol. 3. Cambridge Series in Statistical and Probabilistic Mathematics. Cambridge University Press, Cambridge.

Van Rossum, Guido. 2020. *The Python Library Reference, Release 3.8.2.* Python Software Foundation.

Van Rossum, Guido, and Fred L. Drake. 2009. *Python 3 Reference Manual.* Scotts Valley, CA: CreateSpace.

Vondrick, Carl, Hamed Pirsiavash, and Antonio Torralba. 2016. "Generating Videos with Scene Dynamics." In *Proceedings of the 30th International Conference on Neural Information Processing Systems*, 613–21. NIPS'16. Red Hook, NY, USA: Curran Associates Inc.

Walton, Kendall L. 1993. *Mimesis as Make-Believe: On the Foundations of the Representational Arts.* Harvard University Press.

Wickham, Hadley, Mara Averick, Jennifer Bryan, Winston Chang, Lucy D'Agostino McGowan, Romain François, Garrett Grolemund, et al. 2019. "Welcome to the tidyverse." *Journal of Open Source Software* 4 (43): 1686. https://doi.org/10.21105/joss.01686.

Xu, Lei, Maria Skoularidou, Alfredo Cuesta-Infante, and Kalyan Veeramachaneni. 2019. "Modeling Tabular Data Using Conditional GAN." In *Proceedings of the 33rd International Conference on Neural Information Processing Systems.* Red Hook, NY, USA: Curran Associates Inc.

Yi, Xin, Ekta Walia, and Paul Babyn. 2019. "Generative Adversarial Network in Medical Imaging: A Review." *Medical Image Analysis* 58: 101552. https://doi.org/https://doi.org/10.1016/j.media.2019.101552.

