# OpenReview forum: "Draw Me a Simulator: Using neural networks to build more realistic simulation schemes for causal analysis"
_Computo — Accepted by Computo_

### Review · Reviewer_9BUN · 2025-05-12

**Summary Of Contributions:**

The paper tackles an important and recurring issue. Indeed, simulations are of utmost importance for validation of methodologies, data augmentation, etc. When the data gathered quantitative and qualitative variable and when some information is hidden, it is usually difficult to build a general simulation methodology which really generates data which are close to the reference data.  \\

The paper is clear and well written. The code is well written too. The figures and table well support the text.  I found the more "phylosophical" discussion interesting and usefull.  \\

If I have to point a weakness, I would say that the paper is pretty long. But that's the price you pay for sufficient detail. So I will not ask to shorter it!

**Audience:**

Yes

**Broader Impact Concerns:**

no comment

**Claims And Evidence:**

Yes

**Requested Changes:**

I have some comments and questions (although I think that the paper can be published without corrections) \\

1. I understand why the  the real data are presented pretty late in the paper. I think, that if the application had been presented at the beginning, it would have helped me to get to grips with the chosen example.  \\

2. A paragraph is dedicated to over parametrization. I wonder if the use of some regularization would help to improve the results, expecially for the real data case.  \\

3. In the validation part, three criteria are proposed. I really like the first one for detecting a like "copy-paste" simulation! The second and third ones are also usefull. I wonder if it could be interesting to apply a proper score like the energy score (Gneiting, T.,  Raftery, A. E. (2007). Strictly proper scoring rules, prediction, and estimation. Journal of the American statistical Association, 102(477), 359-378.) to check the simulated joint distribution.

4. In the related works, normalizing flows (D. J. Rezende and S. Mohamed. Variational inference with normalizing flows, 2016.) and Denoising diffusion probabilisticmodels  (Sohl-Dickstein, J., Weiss, E. A., Maheswaranathan, N., and Ganguli, S. Deep unsupervised learning using nonequilibrium thermodynamics, 2015.) could be added to the list.

~

A minor comment :\\
line 308 : $\theta_2 \in \Theta_2$ instead of $\theta_2 \in \Theta$

**Strengths And Weaknesses:**

The paper tackles an important and recurring issue. Indeed, simulations are of utmost importance for validation of methodologies, data augmentation, etc. When the data gathered quantitative and qualitative variable and when some information is hidden, it is usually difficult to build a general simulation methodology which really generates data which are close to the reference data.  \\

The paper is clear and well written. The code is well written too. The figures and table well support the text.  I found the more "phylosophical" discussion interesting and usefull.  \\

If I have to point a weakness, I would say that the paper is pretty long. But that's the price you pay for sufficient detail. So I will not ask to shorter it!

---

> ### Author Response · Authors · 2025-07-16
> **Point-by-point response**
>
> Dear reviewer,
>
> We would like to express our gratitude for your kind and constructive comments.  Below, we provide a point-by-point response to your questions and suggestions.
>
> ## Requested changes
>
> 1. While writing the manuscript, we initially planned to present the real data in Section 2, following the introduction of the running example. However, upon reflection, we decided that postponing Sections 3-5 further would negatively affect the overall flow of the paper.
>
> 2. We did not attempt to implement regularization, though this could easily be incorporated in line with the closing discussion in Section 4.2. Additionally, we would like to point out that, while the models are indeed over-parameterized (in terms of the number of parameters relative to the number of data points used for fitting, and in terms of statistical identifiability), the dimension of the neural network built in Section 5 is relatively low by current standards.
>
> 3. Thank you for pointing out to Gneiting and Raftery (2007). We now refer to it in the conclusion as it may be helpful to derive new criteria to evaluate the quality of a generator.
>
> 4. The two relevant references have been added, thank you.

---

### Review · Reviewer_4PgN · 2025-06-25

**Summary Of Contributions:**

# Summary of the contribution

The paper *Draw Me a Simulator* by Sandrine Boulet and Antoine Chambaz explores an innovative approach to simulation design in causal inference through the use of variational autoencoders (VAEs). The context of causal inference presents particularly
interesting challenges.
The papers contains 8 sections. The first section is an introduction which takes us from the philosophical stance on the use of synthetic data to its mathematical formalization. The second section presents the objectives together with the introduction of the  running example.
Section 3 presents the main notations and a short overview of VAEs and their training difficulties. Section 4 gives details on the construction of VAEs and Section 5 gives details on the implementation of VAE in the specific context of the running example. Section 6 defines the methods to evaluate the performances of a simulator based of VAE. Section 7  is an application / example on real dataset. The paper ends with concluding remarks.

# Conclusion

*Draw Me a Simulator* is a conceptually rich and technically strong contribution to simulation for causal inference. It innovatively combines ideas from machine learning, probability, and philosophy of science to question how define and build simulators based on VAE. This combination of rigorous theory, transparent implementation, and philosophical framing makes this paper both instructive and impactful.

My recommendation is thus to accept this article for publication in *Computo* after correcting a few elements of poorly formatted code and taking into account the comments contained in this report (if these comments are shared by the authors).

**Audience:**

Yes

**Broader Impact Concerns:**

None.

**Claims And Evidence:**

Yes

**Requested Changes:**

# Additional thoughts

Two questions of interest may be discussed in this paper:

First, the computational cost of training such VAEs is not deeply discussed. Insights into runtime, or training stability would be valuable.

Second, a deeper integration with adjacent areas such as synthetic data generation for privacy-preserving analytics or simulation-based inference (SBI) in Bayesian computation would broaden the impact.

# Minor comments

A careful review of the document is recommended to correct a few typos, for example:
- P6 L221: a space is missing after P
- P17 L384: there appear to be problems in the code at this point, but it does not seem to affect the rest of the document.
- P34 L900: The reference is incorrect, producing ??

**Strengths And Weaknesses:**

# General comments

The authors present, with a strong educational focus, the construction of synthetic data using VAE. The inclusion of codes (not verified nor commented in this report) is a real bonus, making this article not only a relevant contribution but also a valuable tool for putting this approach into practice. Given the importance of high-fidelity simulators in evaluating statistical procedures - especially in causal analysis where counterfactuals are central - this work is a very interesting contribution.

The paper is rich in content and well-structured, taking the reader from philosophical reflections to rigorous implementation of VAEs and practical examples. It makes a compelling case for considering machine learning-based simulators as realistic alternatives to classical parametric or bootstrap-based designs.

# Comments on Section 1. Introduction

The philosophical dimension is very interesting and extremely important for this field. The potential for application is undeniable but highly controversial. This ambivalence is evident throughout the document, and all precautions are highlighted to ensure the proper use of these techniques.

# Comments on Section 2. Objective

This section would benefit from a concrete example to better understand the objective of these approaches. This example could also be used to illustrate the different notations used in concrete terms.

It would also be useful to clarify the timing of the observations. For instance in a RCT, it is not clear if covariates refer to baseline covariates or post-randomization covariates. The specific context should be clarified.

# Comments on Section 3. VAE in a nutshell

This is the most theoretical and difficult part of the article. To continue along the educational path we have embarked upon, it would be interesting to clarify what constitutes hyper-parameters (to be chosen by the user) and what constitutes parameters (to be estimated/learned from the data). Furthermore, do the main elements $J_n$, $K$, $\pi$ have a concrete interpretation in a given context? If so, it would be useful to specify.

# Comments on Section 4. How to build a VAE

I appreciate the important choices made by the authors: first, to focus on the “concrete example” rather than presenting a more general framework; second, to address the important issue of overparameterization in a separate subsection.

Another subsection devoted to a discussion on the characteristics of the databases required to allow the construction of a VAE would have been interesting. Especially the question of the volume of data required (Are 5000 "patients" in Section 6. and 3183 in Section 7.  minimal or reasonnable values in practice?). A few words on the importance of dealing with database of quality may be welcome.

# Comments on Section 5. Implementation of the VAE

There are some issues with the code on p. 17 causing several error messages. It would be advisable to resolve the potential issue or at least prevent the error messages from appearing.\\

The rest of this section is clear and particularly well detailed and easy to follow. It would have been preferable to refer to hyper-parameters in order to clarify the text and emphasize the fact that these parameters are chosen by the data scientist in relatinship with medical staff. An explanation of the reasons behind the choices made and the potential impact of these choices on the results would have enriched the content.

# Comments on Section 6. Illustration on simulated data

The simulated experiment is carefully constructed using known functional forms and distributions for treatment, covariates, and outcomes. The goal is not just to fit data, but to build a generator whose output mimics the properties of the data, which is particularly relevant for evaluating inference algorithms in practice. The VAE is trained on this data, and performances is assessed through:
- Visual comparison of empirical distributions (histograms, scatterplots)
- Pointwise comparison of summary statistics (means, variances)
- Predictive accuracy of downstream estimators trained on synthetic vs real data
- Evaluation of the generator’s ability to produce plausible counterfactual outcomes


Particular attention should be paid to the fact that the results in sub-Section 6.2 should be interpreted more as results on the quality of generation rather than as the quality of the generator. This is a very important but complicated point. It would have been interesting to investigate it further, or at least to mention it. The three proposed criteria for evaluating generation (generator) quality are reasonable. To evaluate the generator formalization using divergence measures or information-theoretic tools (e.g., KL, JS, or Wasserstein distance) may be of interest.

The synthetic generator performs well under these criteria, suggesting that the learned mapping captures the main characteristics of the original simulation scheme. Furthermore, the illustration demonstrates that even in moderately high-dimensional settings, VAEs can learn faithful representations of complex joint distributions.

Let me point that the introduction of DAG in this context is very interesting even inspiring (Figure 4).

# Comments on Section 7. Illustration on real data

This section presents an empirical evaluation of the proposed method using real-world data from the International Warfarin Pharmacogenetics Consortium, a widely used benchmark in personalized medicine. Here, the focus is on the dosage prediction problem, which involves patient-specific covariates and treatment outcomes.

What is the timing of variables A and Y recording (measurement)? This should be specified in the data description.

# Comments on Section 8. Conclusion

Section 8 contains some very interesting information, including a list of related works and some elements on the philosophical approach to the question. The section also mentions certain limitations presented as challenges for future research. In my opinion, subsection 8.2 would benefit from a very clear summary of the results/hypotheses/constraints encountered during the construction of the VAE as a "Take Home Message" for the reader.

While the VAE is still able to produce plausible samples, the gap between real and generated data is more apparent. Challenges such as covariate shift, missing confounders, or poor support overlap emerge.
This is a valuable insight: real-world data often violate the idealized assumptions of simulation. Nonetheless, the authors remains optimistic and treats these limitations not as failures, but as indicators of future work - particularly the need to embed causal constraints directly into neural architectures. It is a relevant message.

---

> ### Author Response · Authors · 2025-07-16
> **Point-by-point response**
>
> Dear reviewer,
> Thank you for your kind and constructive comments.  Below, we provide a point-by-point response to your questions and suggestions.
> ## Section 2
> - We added a few lines to the presentation of the running example in section 2 (not shown here for lack of space).
> ## Section 3
> - We added this sentence at the end of the first paragraph of section 3 (not shown entirely for lack of space):
>   > Moreover, we emphasize that the entire parameter $\theta$ is to be fitted (...)
> - Interpretations for $J_n$ and $K$ are already provided in the enumerated list at the end of section 3.1: $J_n$ is used to sample one observation, say $O_i$, uniformly among all genuine observations; $K$ samples a neighbor of the encoded version of $O_i$ in the latent space.  As for $\pi$, its interpretation is given in the first paragraph of section 2, where it is described as the mapping that projects a complete data set onto a coarser real-world data. We simply added a sentence to item 3 (Gaussian sampling) :
>   > In other words, $K$ enables the sampling of a neighbor of the encoded version of $O_i$ in the latent space.
> ## Section 4
> - The question of what volume of data is required is indeed an important one. It is widely accepted that "the more data, the better." However, this simple adage overlooks the need to explicitly link the required data volume to both (a) the size and complexity of the input data, and (b) the complexity of the neural network being fitted. In this study, we chose to (a) simulate five times as many genuine observations as the number of observations in the IWPC real data example, and (b) build neural networks of comparable complexity in both illustrations (see sections 6, 7). When testing our approach to building a simulator on simulated data, we found that increasing the sample size of the genuine observations by two- to threefold, and moderately augmenting the complexity of the neural network (e.g., by adding one or two more layers to the encoder and each component of the decoder), did not lead to significant changes in the quality of the results.
> - The quality of the data is indeed of paramount importance, another factor overlooked by the adage.  However, we did not investigate this aspect, as demonstrated by the absence of missing values in both illustrations.
> - These two comments now appear in the article's conclusion (not shown here for lack of space).
> ## Section 5
> - We did not manage to remove the error message, which does not appear on our laptops.
> -  The choice of architecture is, in a sense, a meta-hyperparameter.  In addition, several other hyperparameters must be selected by the data scientist, including the number of layers, the number of neurons per layer, $\beta$, and the parameters used in Algorithm 1.  It is certain that the architecture can be improved on a case-by-case basis.  The data scientist's task could benefit from input from domain experts.  As for the other hyperparameters, we do not claim that our choices are optimal. However, selecting appropriate optimality criteria and optimizing the choices based on them are challenging tasks.
> The comment now appears in the article's conclusion.
> ## Section 6
> - The reviewer writes "[T]he results in sub-Section 6.2 should be interpreted more as results on the quality of generation rather than as the quality of the generator.  This is a very important but complicated point."  This is an excellent observation.
> Ideally, we would like to evaluate the proximity between the generator associated with $\Gen_{\theta_n}$ and the true law $P$ of the observations.  However, in practice, we do not have access to either $\law(\Gen_{\theta_n}(Z))$ or $P$.  One approach would be to estimate a measure of discrepancy between these two laws (e.g., Kullback-Leibler divergence) using data sampled from each, which would allow us to test whether one of several generators is closer to the true law than the others. However, we have opted for a different approach.
> We added the above paragraph to the beginning of sub-Section 6.2.
> ## Section 7
> - The reviewer raises a valid point. We confirm that $A$ precedes $Y$ based on our choices, and this has been clarified in the text.
> ## Section 8
> - We added a take-home message at the end of sub-Section 8.2 (not shown here for lack of space)
> ## Additional thoughts
> - We now report that the training loop takes about 10 minutes to run on the simulated data. It takes as much time to run the training loop on the real data.  We did not rigorously investigate the stability issue.  However, stability has never been problematic throughout the development of this project.
> - Thank you for pointing out the links to privacy-preserving analytics and simulation-based inference. We now highlight the potential relevance of our study for privacy-preserving analytics, while touching on the issue of simulator sharing in sub-Section 8 and the new take-home message.
> ## Minor comments
> - Thank you for pointing out these typos. They have been corrected.

---

### Comment · Action_Editor_z6xN · 2025-06-25
**Start of rebuttal period**

Dear authors,

We have received two reports for your submission to Computo entitled “Draw Me a Simulator: Using neural networks to build more realistic simulation schemes for causal analysis”.

A period of 6 weeks is allowed for discussion with the referees before they issue a final opinion. During this period, you can make any changes to your submission that you feel are necessary and that you are able to make. At the end of this period, a decision will be made, ranging from final acceptance to more substantial requests for modification.

Best regards

---

> ### Author Response · Authors · 2025-07-16
> **Revision carried out**
>
> Dear Editor,
>
> We would like to express our gratitude to the reviewers for their kind and constructive comments. Below, we provide a point-by-point response to each of their questions and suggestions.
>
> Kind regards, Antoine Chambaz & Sandrine Boulet
>
> PS: At the moment, the article does not compile properly on the journal's website (although it does on our laptops). We've contacted François-David Collin to help us solve the issue (the construction of the virtual machine fails). We can share the compiled PDF file.

---

> > ### Comment · Action_Editor_z6xN · 2025-07-17
> > **PDF version of revised manuscript**
> >
> > Dear reviewers,
> >
> > While we are working on solving the compilation issue mentioned by the authors, you fill find the pdf version of the revised manuscript at this url:
> >
> > https://plmbox.math.cnrs.fr/seafhttp/f/f38a54968e484e9d89b6/?op=view

---

### Comment · Action_Editor_z6xN · 2025-07-21
**End of rebuttal period: reviewers should submit their official recommendation**

Dear reviewers,

The authors have responded to your comments and produced an updated version of their manuscript, available here :

- https://github.com/achambaz/draw_me_a_simulator (github repo)
- https://plmbox.math.cnrs.fr/seafhttp/f/f38a54968e484e9d89b6/?op=view (PDF manuscript)

You can now make a recommendation based on the authors' responses and the updated version of the submission. On the basis of your recommendations, the editorial board will make a decision (acceptance, major or minor revisions, rejection).

Because of the summer period, we have postponed the deadline to August 31, but you are of course very welcome to submit your recommendation sooner if you can!

Thank you for your time in reviewing this manuscript.

---

> ### Comment · Reviewer_9BUN · 2025-07-21
> **accept**
>
> I think that the paper is now ready for acceptation.

---

> ### Comment · Reviewer_4PgN · 2025-08-18
> **In favor of publishing this article**
>
> Convinced by the modifications and additions/corrections made to the previous version of the manuscript, I am now in favor of publishing this article.

---

### Note · Reviewer_9BUN · 2025-07-21

**Comment:**

It is a very nice paper. The authors replied to all the reviewer's comment.

**Audience:**

Yes

**Claims And Evidence:**

Yes

**Decision Recommendation:**

Accept

---

### Note · Reviewer_4PgN · 2025-08-25

**Comment:**

The results presented in this paper are of major interest. The paper is well written specifying messages of importance.

**Audience:**

Yes

**Claims And Evidence:**

Yes

**Decision Recommendation:**

Accept

---

### Decision · Action_Editor_z6xN · 2025-09-04

**Recommendation:** Accept as is

**Comment:**

Both reviewers are satisfied that their comments have been taken into account by the authors.

**Audience:**

Fit to Computo's scope

**Claims And Evidence:**

In agreement with the elements provided by the reviewers, the claims and evidence are sufficiently solid for publication.

---

> ### Decision · Editors_In_Chief · 2025-09-08
>
> I approve the AE's decision.